EMBO
Molecular Medicine

# Targeting YAP to overcome acquired resistance to ALK inhibitors in *ALK*-rearranged lung cancer

Mi Ran Yun[1,2,†] (iD), Hun Mi Choi[2,†], You Won Lee[2], Hyeong Seok Joo[1], Chae Won Park[2], Jae Woo Choi[3,4], Dong Hwi Kim[1], Han Na Kang[1,2], Kyoung-Ho Pyo[2], Eun Joo Shin[2], Hyo Sup Shim[5], Ross A Soo[6], James Chih-Hsin Yang[7,8], Sung Sook Lee[9], Hyun Chang[10], Min Hwan Kim[2], Min Hee Hong[2], Hye Ryun Kim[2] & Byoung Chul Cho[2,*] (iD)

## Abstract

Clinical benefit of ALK tyrosine kinase inhibitors (ALK-TKIs) in *ALK*-rearranged lung cancer has been limited by the inevitable development of acquired resistance, and bypass-molecular resistance mechanisms remain poorly understood. We investigated a novel therapeutic target through screening FDA-approved drugs in ALK-TKI-resistant models. Cerivastatin, the rate-limiting enzyme inhibitor of the mevalonate pathway, showed anti-cancer activity against ALK-TKI resistance *in vitro/in vivo*, accompanied by cytoplasmic retention and subsequent inactivation of transcriptional co-regulator YAP. The marked induction of YAP-targeted oncogenes (EGFR, AXL, CYR61, and TGFβR2) in resistant cells was abolished by cerivastatin. YAP silencing suppressed tumor growth in resistant cells, patient-derived xenografts, and *EML4-ALK* transgenic mice, whereas YAP overexpression decreased the responsiveness of parental cells to ALK inhibitor. In matched patient samples before/after ALK inhibitor treatment, nuclear accumulation of YAP was mainly detected in post-treatment samples. High expression of YAP in pretreatment samples was correlated with poor response to ALK-TKIs. Our findings highlight a crucial role of YAP in ALK-TKI resistance and provide a rationale for targeting YAP as a potential treatment option for *ALK*-rearranged patients with acquired resistance to ALK inhibitors.

**Keywords** acquired resistance; ALK; non-small cell lung cancer; statin; YAP
**Subject Categories** Cancer; Pharmacology & Drug Discovery; Respiratory System

## Introduction

Rearrangement of the anaplastic lymphoma kinase (*ALK*) gene defines a distinct clinicopathologic subset of non-small cell lung cancer (NSCLC) and occurs in approximately 3–7% of NSCLC cases, with a higher prevalence among younger, never/light smoker, and adenocarcinoma histology (Shaw *et al*, 2009). Crizotinib is currently the standard first-line therapy for patients with advanced ALK-positive NSCLC (Solomon *et al*, 2014). However, most patients who initially respond to crizotinib invariably relapse within 1 year due to the emergence of drug resistance (Katayama *et al*, 2015). Recently, more potent second-generation ALK tyrosine kinase inhibitors (TKIs) have been developed for the treatment of crizotinib-resistant and crizotinib-naïve patients, but acquired resistance to these agents is also an inevitable problem (Awad & Shaw, 2014). Based on published reports, the mechanisms of resistance to ALK-TKIs can be broadly classified as ALK-dependent alterations (e.g., ALK secondary mutations and ALK gene amplification) and ALK-independent alterations (e.g., upregulation of bypass signaling pathways; lineage changes; and drug efflux pump) (Katayama *et al*, 2012; Lin *et al*, 2017). ALK-dependent mechanisms generally indicate continued dependency on ALK signaling and potential sensitivity to other ALK-TKIs, depending on the type of secondary mutation. In a recent report by Gainor *et al* (2016), resistance mutations were found in 20 and 50% of patients following treatment with crizotinib and the second-generation ALK-TKIs (e.g., ceritinib and alectinib), respectively. This indicates that at least half of patients exhibit ALK-independent mechanisms upon acquisition of acquired resistance to ALK-TKIs. Several examples of bypass signaling activation have been proposed (Crystal *et al*, 2014; Hrustanovic *et al*,

---

1   JEUK Institute for Cancer Research, JEUK Co., Ltd., Gumi-City, Korea
2   Division of Medical Oncology, Yonsei Cancer Center, Yonsei University College of Medicine, Seoul, Korea
3   Department of Pharmacology, Yonsei University College of Medicine, Seoul, Korea
4   Severance Biomedical Science Institute, Yonsei University College of Medicine, Seoul, Korea
5   Department of Pathology, Yonsei University College of Medicine, Seoul, Korea
6   Department of Haematology-Oncology, National University Cancer Institute, Singapore, Singapore
7   Graduate Institute of Oncology, National Taiwan University, Taipei, Taiwan
8   Department of Oncology, National Taiwan University Hospital, Taipei, Taiwan
9   Department of Hematology-Oncology, Inje University Haeundae Paik Hospital, Busan, Korea
10  International St. Mary's Hospital, College of Medicine, Catholic Kwandong University, Incheon, Korea
    *Corresponding author. Tel: +82 2 2228 0870; Fax: +82 2 393 3562; E-mail: cbc1971@yuhs.ac
    †These authors contributed equally to this work as first/second authors

2015), but major mechanisms to explain widespread emergence of resistance are still unknown.

Yes-associated protein (YAP), a major downstream effector of the Hippo pathway (Maugeri-Saccà & De Maria, 2018), serves as a transcriptional regulator by facilitating transcription of pro-growth genes and suppressing pro-apoptotic genes. YAP has emerged as a critical oncogene in multiple cancer types (Zanconato et al, 2016; Maugeri-Saccà & De Maria, 2018). In particular, YAP has been reported to be closely associated with the emergence of resistance to conventional chemotherapeutics (Gujral & Kirschner, 2017) and BRAF/MEK/EGFR-targeted therapies (Lin et al, 2015; Lee et al, 2018). However, only a few studies have verified the clinical relevance of YAP activation in human samples.

The mevalonate (MVA) pathway is an essential metabolic pathway that produces sterols and isoprenoids that are essential for tumor growth and progression (Mullen et al, 2016). MVA pathway metabolite isoprenoids, such as farnesyl pyrophosphate (FPP) and geranylgeranyl pyrophosphate (GGPP), are necessary for the post-translational prenylation of small GTPases that play a significant role in multiple intracellular signaling pathways. A number of studies have shown that statins, which inhibit the rate-limiting enzyme of the MVA pathway, 3-hydroxy-3-methyl-glutaryl-CoA reductase (HMGCR), exhibit anti-cancer effects in many cancers by reducing isoprenoids (Yu et al, 2018). Notably, GGPP has recently been found to activate YAP by inhibiting its phosphorylation and promoting its nuclear accumulation. With this mechanism, statins as antagonists of isoprenylation can inhibit YAP nuclear localization and transcriptional responses, leading to growth arrest and apoptosis of cancer cells (Zhao et al, 2008; Zhang et al, 2009; Sorrentino et al, 2014; Wang et al, 2014; Mi et al, 2015).

This study aims to identify novel resistance mechanisms to ALK-TKIs by utilizing high-throughput screening using a library of FDA-approved drugs composed of a collection of 640 clinically used compounds with known and well-characterized bioactivity, safety, and bioavailability. Here, we report a previously unidentified YAP-mediated mechanism of acquired resistance to crizotinib in ALK-positive NSCLC using in vitro and in vivo models with subsequent validation in patient samples before and/or after ALK inhibitor therapy. Ultimately, our findings provide a novel promising therapeutic strategy targeting YAP signaling to overcome acquired resistance to ALK-TKIs in ALK-positive NSCLC.

# Results

### Statins exhibit *in vitro* and *in vivo* anti-cancer activity against crizotinib-resistant cells

We generated crizotinib-resistant cells (CR cells; CR pool, CR #1 and CR #3) as described in the Materials and Methods. These CR cells exhibited lower phosphorylated and total ALK levels concomitant with morphological changes from round to fibroblast-like cells compared with that of parental cells (Appendix Fig S1A–C). Silencing ALK using small interfering RNA (siRNA) transfection and ALK inhibitors ceritinib and lorlatinib had no effect on the growth of CR cells (Appendix Fig S1D and E).

Moreover, sequencing of the ALK tyrosine kinase domain of resistant cells showed no secondary ALK mutations. Altogether, CR cells were unlikely to have arisen by ALK-dependent mechanisms.

To uncover novel signaling pathways related to crizotinib-acquired resistance, we screened a 640 FDA-approved drug library for drug efficacy in parental and CR pool cells. The average z-score was calculated for drug effect on cell viability. Cerivastatin was the most selective hit for CR pool cells compared with parental cells, and further confirmed a greater decrement in cell viability of other CR cells (CR #1 and CR #3; Fig 1A–C). Considering that cerivastatin was withdrawn from the market in 2001 due to fatal rhabdomyolysis and kidney failure (Furberg & Pitt, 2001), we additionally evaluated the anti-proliferative capability of the other clinically available statins, atorvastatin and simvastatin, in CR cells. Although they were used at higher concentrations than cerivastatin, these statins successfully inhibited cell growth at the concentration range reported in other previous preclinical studies (Fig EV1A and Appendix Fig S2A). Cerivastatin, simvastatin, and atorvastatin remarkably increased the expression levels of c-PARP, c-Cas3, and p21 in CR cells (Fig EV1B, and Appendix Fig S2B). These in vitro findings were further confirmed by in vivo xenograft studies showing that cerivastatin and atorvastatin significantly delayed tumor growth of the CR pool (Figs 1D and EV1C). Based on the anti-cancer effects of statins, cerivastatin with the lowest $IC_{50}$ was used as a representative in subsequent experiments despite being a clinically discontinued drug.

Statins are a class of cholesterol-lowering drugs that reduce cardiovascular diseases by blocking HMGCR, a rate-controlling enzyme of the MVA pathway (Appendix Fig S3) (Mullen et al, 2016; Iannelli et al, 2018). Among the important metabolites of the MVA pathway, the addition of MVA and GGPP was able to rescue cerivastatin-induced cell growth inhibition in CR cells, whereas FPP and squalene had no effect (Fig 1E). Addition of GGPP markedly abrogated cerivastatin- and atorvastatin-induced c-PARP, c-Cas3, and p21 expression in CR pool, but not in parental cells (Figs 1F and EV1D). Conversely, geranylgeranyl-transferase I inhibitor GGTI-298 was able to reproduce the effects of statin by increasing the expression of c-PARP, c-Cas3, and p21 in CR cells in a dose-dependent manner (Fig 1F). However, farnesyl transferase inhibitor FTI-277 only weakly increased p21 expression and we failed to detect c-PARP and c-Cas3 levels (Appendix Fig S4).

Reportedly, the MVA pathway is associated with mutant p53 expression in a variety of cancer types (Freed-Pastor et al, 2012; Sorrentino et al, 2014; Parrales et al, 2016; Turrell et al, 2017). However, there was no difference in the p53 mutational status between H3122 parental and resistant cells (Appendix Table S1). These results suggest that anti-cancer activity of statin in resistant cells may be independent of TP53 mutation.

### Cerivastatin decreases upregulation of multiple oncogenic factors by inactivating YAP

Given that the MVA-GGPP axis is involved in cancer progression by activating the transcriptional co-regulator YAP as a Hippo pathway downstream effector (Zhao et al, 2008; Zhang et al,

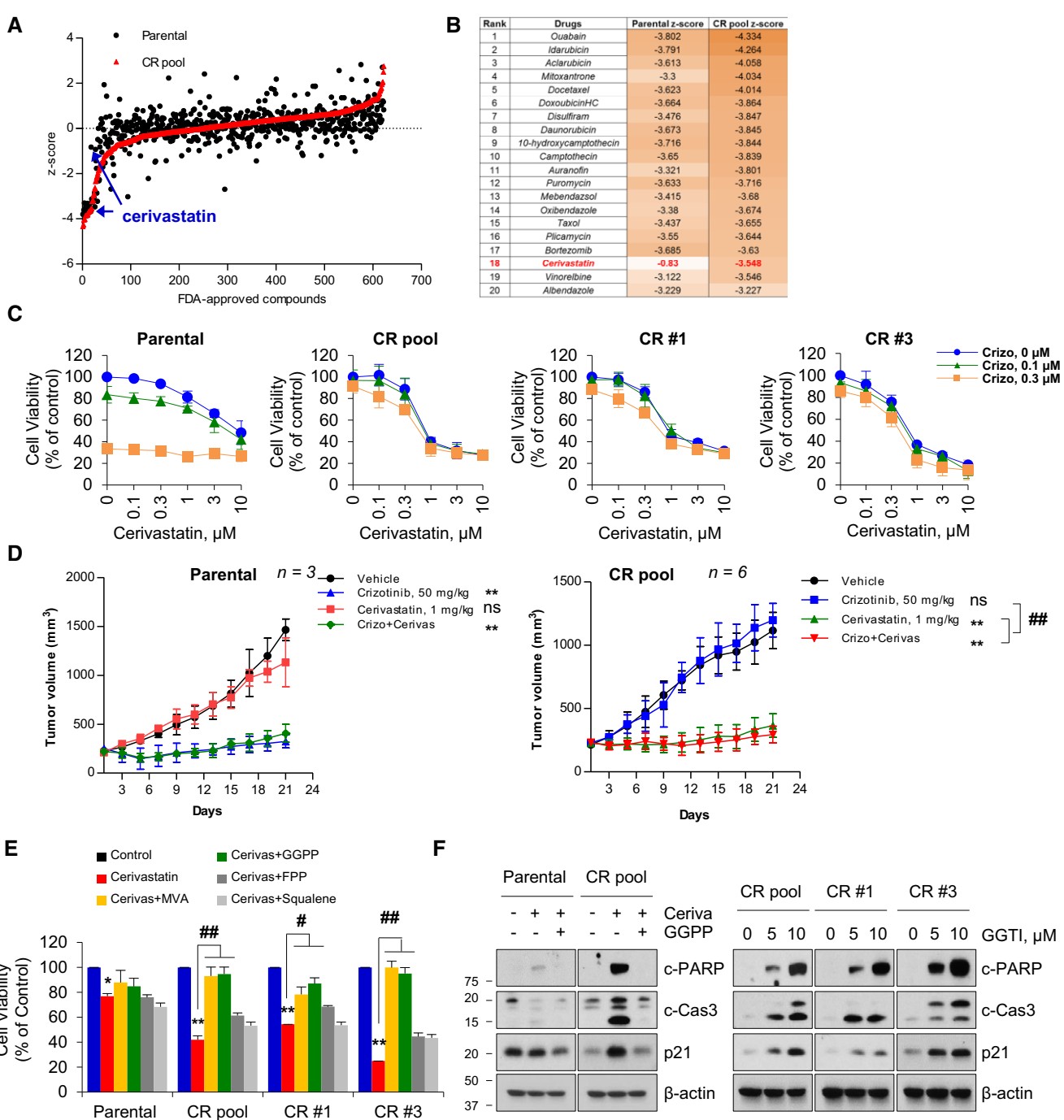

**Figure 1. *In vitro* and *in vivo* anti-cancer activity of cerivastatin against CR cells.**

A   Z-scores for FDA-approved compounds on cell viability of parental (black circle) and CR pool (red triangle) cells.

B   Top 20 list of identified hits in the screen of FDA-approved drugs.

C   Cell viability curves of parental and CR cells in response to combined treatment with cerivastatin and crizotinib (n = 3).

D   Tumor growth curves of parental (n = 3) and CR pool xenografts (n = 6) treated with the indicated drugs (Kruskal–Wallis with Dunn's *post hoc* test: **P < 0.01 vs. vehicle, ##P < 0.01 vs. crizotinib treatment. ns, not significant).

E   Bar chart showing cell viability of cells after cerivastatin treatment in the presence and absence of metabolites of the mevalonate pathway. MVA: mevalonic acid (0.5 mM), GGPP: geranylgeranyl pyrophosphate (10 μM), FPP: farnesyl pyrophosphate (10 μM), squalene (10 μM; ANOVA with Tukey's *post hoc* test: *P < 0.05, **P < 0.01 vs. control in each cell. #P < 0.05, ##P < 0.01 vs. the value at the indicated comparison in each cell; n = 4).

F   Representative immunoblots of the indicated proteins in cells treated with cerivastatin (1 μM) alone or with GGPP (10 μM) for 24 h (left) and in lysates of cells treated with GGTI-298 for 24 h (right). Blots are representative of three independent experiments.

Data information: Data represent means ± SD (C and D) or ± SEM (E).
Source data are available online for this figure.

2009; Sorrentino *et al*, 2014; Wang *et al*, 2014; Mi *et al*, 2015), we investigated the relevance of the YAP signaling pathway in crizotinib resistance. Figure 2A showed that CR pool cells exhibited a lower level of basal YAP phosphorylation and a significant nuclear accumulation of YAP compared with that of parental cells, indicating an increase in the transcriptional activity of YAP. Treatment with cerivastatin and atorvastatin (Figs 2A and EV1E) resulted in a robust increase in YAP phosphorylation and translocation of nuclear YAP into the cytoplasm, which was reversed by the addition of GGPP. Despite depletion of endogenous expression of the Hippo pathway core component LATS1/2 by siRNA, moreover, YAP was still phosphorylated upon treatment with cerivastatin, and dephosphorylated after the addition of GGPP (Appendix Fig S5). These results suggest that cerivastatin inhibits GGPP-mediated YAP activation in a Hippo pathway-independent manner.

Next, we found that the CR pool cells exhibit a significant upregulation of YAP target gene signature compared with that of parental cells (Fig 2B and Appendix Fig S6). Based on functional enrichment analysis, interaction network analysis between YAP and genes belonging to the top 15 signaling pathways including metabolic pathway, focal adhesion, cytoskeleton pathways, and Ras signaling demonstrated that a number of genes were functionally connected to YAP (Fig 2C). In particular, we focused on EGFR, AXL, and TGFβR2, which are known to be involved in drug resistance (Miyawaki *et al*, 2017; Yun *et al*, 2018), as well as CYR61, which is a direct YAP target. Upregulation of these genes in CR pool cells was confirmed by Western blot analysis (Fig 2D). We also found that the expression levels of vimentin (VIM), cyclin D1, insulin-like growth factor-binding protein 3 (IGFBP3), and the MAPK phosphatase DUSP6 varied depending on the CR clones, indicating heterogeneity in the resistant clones (Appendix Fig S7A). YAP inhibition with siRNAs markedly suppressed the expression of EGFR, AXL, TGFβR2, and CYR61, but had no effect on expression levels of the epithelial-to-mesenchymal transition (EMT)-associated gene VIM and tumor suppressor genes IGFBP3 and DUSP6 (Fig 2E and Appendix Fig S7B). Verteporfin (VP), a pharmacological inhibitor of YAP, is known to inhibit YAP transcriptional activity by preventing the interaction of YAP and TEA domain family members (Liu-Chittenden *et al*, 2012; Yu *et al*, 2014). Treatment with VP, cerivastatin, or GGTI-298 also showed similar results (Fig 2F). Given that YAP and its paralogue transcriptional co-activator with PDZ-binding motif (TAZ) are known to be functionally redundant and similarly regulated by Hippo signaling (Zhang *et al*, 2009; Moroishi *et al*,

2015), we further determined the role of TAZ in our system. Unlike YAP, TAZ knockdown did not affect the expression of EGFR, AXL, and TGFβR2, but robustly inhibited CYR61 and VIM expression and increased IGFBP3 expression (Fig EV2 and Appendix Fig S7). These results suggest that there are potential differences between YAP and TAZ in the context of ALK-TKI resistance.

## YAP activation promotes resistance to ALK-TKI *in vitro* and *in vivo*

We next examined whether YAP, the target of cerivastatin identified from the drug screen, is a novel target of ALK resistance. To this end, we established the following three stable cell lines from H3122 cell line: (i) endogenous YAP-expressing pLVX cell line as control, (ii) YAP-WT cell line overexpressing wild-type YAP, and (iii) YAP-S127A cell line overexpressing constitutively active YAP. Immunoblot analysis demonstrated that YAP was successfully overexpressed in both YAP-WT and YAP-S127A cells. Phosphorylated levels of YAP in YAP-S127A cells were the lowest among the three stable cell lines (Fig EV3A). YAP-WT and YAP-S127A cells exhibited markedly reduced susceptibility to crizotinib concomitant with upregulation of CYR61, EGFR, and AXL expression (Fig EV3A and B).

To clarify the functional role of YAP in ALK-TKI resistance *in vivo*, pLVX, YAP-WT, and YAP-S127A cells were injected subcutaneously into nude mice. Tumor xenografts derived from YAP-WT and YAP-S127A cells grew faster than those from pLVX cells. On day 21 of tumor growth, tumor volumes of the YAP-S127A xenograft were the largest, indicating a potential role of active YAP on tumor progression (Fig 3A). Consistent with the *in vitro* findings, antitumor efficacy of crizotinib was remarkably reduced in both YAP-WT and YAP-S127A tumors. Treatment with cerivastatin significantly suppressed tumor growth in YAP-WT ($P < 0.05$ vs. vehicle) and YAP-S127A xenografts ($P < 0.05$ vs. vehicle), but not in pLVX xenograft (Fig 3B). Notably, the combined treatment of cerivastatin and crizotinib produced a potent anti-tumor synergy compared with that of crizotinib ($P < 0.05$) or cerivastatin alone ($P < 0.05$) in YAP-WT xenograft. Interestingly, YAP-S127A xenograft was less responsive to cerivastatin compared with YAP-WT xenograft. Moreover, combining of cerivastatin with crizotinib was comparable to the effect of single-agent treatment. These results may imply a limited inhibitory effect of cerivastatin on artificial YAP activity because YAP-S127A is not due to GGPP-mediated YAP activity. In line with

---

**Figure 2. GGPP-mediated yes-associated protein (YAP) activation.**

A  Left: representative immunoblots for the indicated proteins in lysates of cells treated with cerivastatin (1 µM) alone or with GGPP (10 µM) for 24 h. Middle: immunofluorescent staining of YAP (red) and DAPI (blue) in cells treated with the method described on the left. Right: representative immunoblots for the indicated proteins in nucleus/cytoplasm fractionation of cells treated with the same conditions as on the left.

B  Microarray analysis heat-map showing a published YAP signature in CR pool cells compared with parental cells ($P < 0.05$ and fold change > 1.5-fold). The data set is provided in Table EV1.

C  Interaction network analysis between YAP and genes belonging to the top 15 categories using the Cytoscape program. Major neighbor genes that are connected with YAP are highlighted in circle. The red circle refers to upregulated genes, and the green circle refers to downregulated genes.

D, E  Representative immunoblots of the indicated proteins in basal lysates of CR cells compared with parental cells (D) and in lysates of CR cells transfected with either negative control siRNA (Con si) or YAP siRNAs (two sets of siRNAs against YAP: YAP si#1 and YAP si#2) (E).

F  Representative immunoblots of the indicated proteins in CR cells after treatment of VP, cerivastatin, or GGTI-298 for 24 h.

Data information: Blots are representative of three independent experiments.
Source data are available online for this figure.

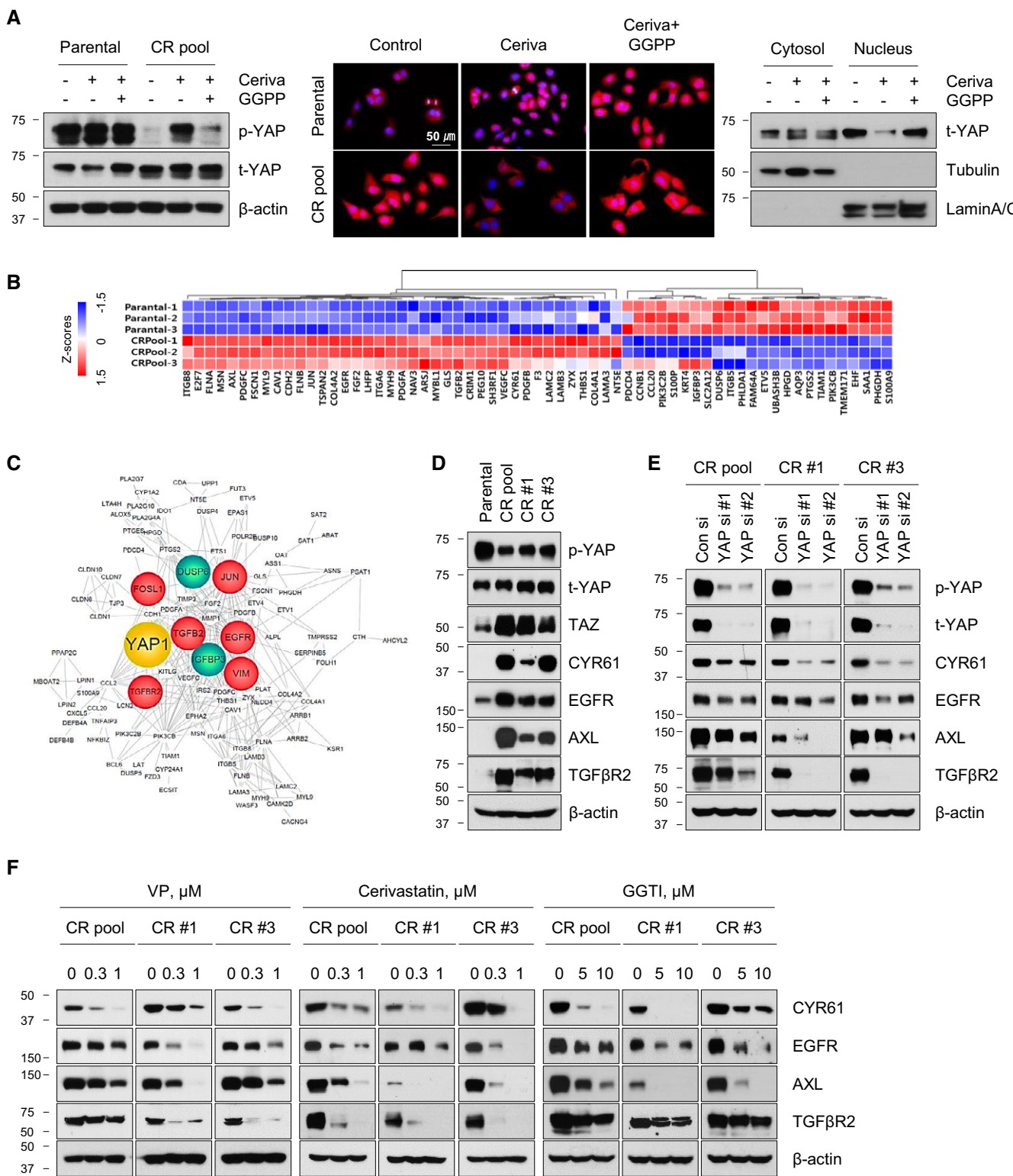

**Figure 2.**

these results, upon combination of cerivastatin and crizotinib, overall expression and nuclear localization of YAP were completely suppressed in tumor sections of YAP-WT, but nuclear YAP subpopulation was still detected in that of YAP-S127A (Fig 3C). Taken together, although both YAP overexpression and activation are responsible for resistance to crizotinib, transcriptional activity of YAP may be more aggressive and play a critical role in tumor progression.

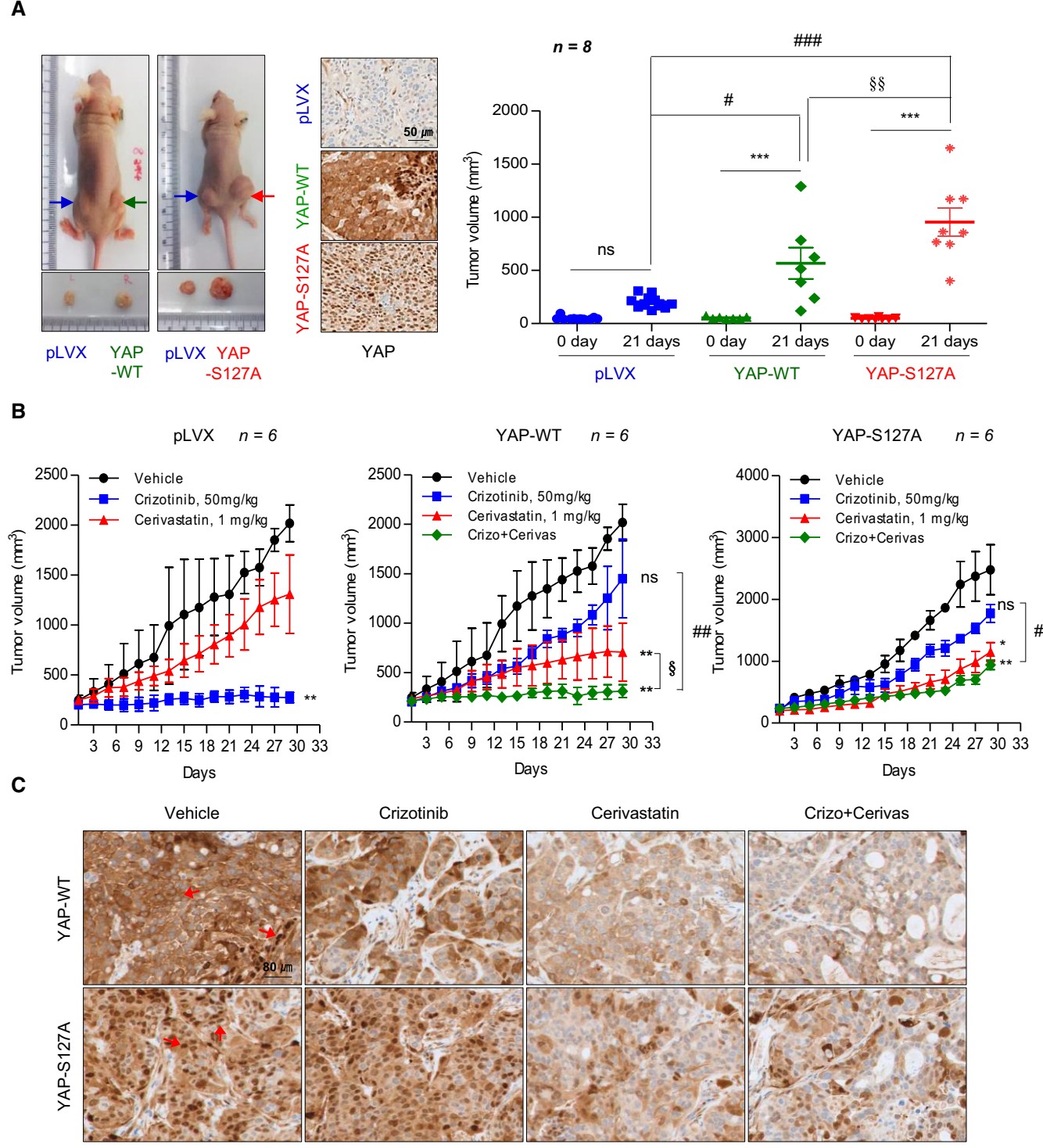

**Figure 3. Effect of yes-associated protein (YAP) overexpression on the response to crizotinib.**

A   Left: representative images showing tumorigenesis of nude mice implanted with pLVX (left flank) and YAP-WT or YAP-S127A (right flank) stable cells. Middle: representative images of IHC of YAP in sections from the indicated xenograft tumor. Right: comparison of xenograft tumor volumes after 3 weeks from when tumor volume reached 40–50 mm$^3$ on the left. ***$P < 0.001$ vs. the value on day 0 in each group. #$P < 0.05$, ###$P < 0.001$ vs. the value for the indicated comparison §§$P < 0.01$ vs. the value on day 21 in YAP-WT. ns, not significant.

B   Tumor growth curves of pLVX, YAP-WT, and YAP-S127A xenografts during treatment with crizotinib (50 mg/kg), cerivastatin (1 mg/kg), or a combination of the two. *$P < 0.05$, **$P < 0.01$ treatment group vs. vehicle in each xenograft. #$P < 0.05$, ##$P < 0.01$, §$P < 0.05$ vs. the value for the indicated comparison. ns, not significant.

C   Representative IHC images of YAP staining in tumor sections from mice treated as indicated. The red arrow indicates YAP staining.

Data information: Data represent means ± SD (A and B). Kruskal–Wallis with Dunn's *post hoc* test was used for comparing multiple groups.

**Inhibition of YAP overcomes tumor sensitivity to ALK-TKIs in mouse xenografts, patient-derived xenograft models, and *EML4-ALK* transgenic mice**

The average *z*-score of YAP short hairpin RNA (shRNA) was higher than that of other genes in shRNA screen of 1,000 genes on the survival of multiple ALK-TKI-resistant patient-derived cells (PDCs) in publicly available data sets (Dardaei *et al*, 2018; Appendix Fig S8), supporting our findings that YAP is closely associated with ALK-TKI resistance. To verify the potential of YAP as a promising target overcoming resistance to ALK-TKI, we evaluated the effect of YAP inhibition by siRNA/shRNA in ALK-TKI-acquired-resistant *in vitro*/*in vivo* models. YAP silencing markedly reduced the proliferation and clonogenicity of CR cells mainly due to cell cycle arrest at G0/G1 phase with induction of p21 expression, which was slightly enhanced in co-treatment with crizotinib (Figs 4A and B, and EV4). Similar results were obtained with ceritinib-acquired-resistant cells (LR pool and LR #6) displaying higher expression of YAP and YAP target genes compared with that of parental cells (Appendix Fig S9). In contrast, TAZ silencing failed to attenuate the clonogenicity of resistant cells, except for CR #3 cells (Appendix Figs S9 and S10). In xenograft models, following subcutaneous cell injection, tumors from control cell were mostly observed within 2 weeks, but those from stable YAP knockdown cells began to appear in about 1 month and were consequently smaller at the end of the experiment (Fig 4C). In line with results, a YAP pharmacological inhibitor VP treatment yielded superior tumor growth inhibition (TGI) compared with vehicle in CR pool xenograft (Fig 4D). Considering that VP has been clinically used as a photosensitizer in photodynamic therapy (Bressler & Bressler, 2000; Battaglia Parodi *et al*, 2016), our results showed that VP exhibits significant therapeutic effects against ALK-TKI resistance by inhibiting YAP transcriptional activity without light activation, which is consistent with other reports (Brodowska *et al*, 2014; Slemmons *et al*, 2015; Cheng *et al*, 2016; Ma *et al*, 2016). The *in vivo* activity of YAP inhibition was further validated in crizotinib-acquired-resistant patient-derived xenograft (PDX) models (YHIM-1001CR) exhibiting predominant nuclear accumulation of YAP protein (Fig 5A and Appendix Fig S12). Figure 5B showed a significant nuclear accumulation and overexpression of YAP in progressive disease (PD) on crizotinib or ceritinib compared with control in *EML4-ALK* transgenic mouse model. Following PD on ceritinib treatment, combined treatment with ceritinib and VP led to pronounced tumor shrinkage and complete remission after 2 weeks, whereas continued treatment with ceritinib alone led to further growth of the lung nodules (Fig 5C).

Taken together, these results demonstrate that targeting YAP is a potential therapeutic option for resistance of ALK-TKIs *in vitro* and *in vivo*.

**Clinical implications of nuclear YAP expression in tumor biopsies from patients with ALK rearrangement**

To explore the clinical relevance of our findings, we assessed the expression pattern of YAP in tumor biopsies acquired before and/or after ALK inhibitor therapy from 17 ALK-positive patients (Table 1). All 17 patients had been treated with ALK inhibitors (crizotinib or ceritinib), and both pre- and post-treatment samples were available for nine of these patients. The intensity of overall YAP expression was not significantly different between pre- and post-treatment tumors, but nuclear YAP staining was more prominent in post-treatment tumors compared with pretreatment tumors (Fig 6A and B). Next, to determine whether the degree of YAP expression is associated with poor prognosis, the origin patients of tumor biopsies were classified into partial response (PR) and stable disease (SD)/PD groups according to the best response to ALK-TKI, after which YAP staining intensity was compared between pretreatment tumor biopsies of the two groups (Fig 6C). Of the 10 PR patients, 9 (90%) showed a staining intensity score of 1 (low) to 2 (intermediate) for YAP. On the other hand, of 5 SD/PD patients, 3 (60%) showed a staining intensity score of 3 (high) for YAP. These results indicate that high YAP expression may affect efficacy to ALK inhibitors. Meanwhile, we then investigated genetic mutation status of the seven biopsy samples available by targeted sequencing (Fig EV5, Table EV2, and Appendix Table S2) and found no genetic alterations in YAP itself or in its signaling pathway. Only 1 sample (14.3%) revealed an L1196M mutation (Awad & Shaw, 2014; Gainor *et al*, 2016), which is the most commonly known ALK secondary mutation, and showed relatively low YAP expression. Moreover, several TP53 mutations (P33R, E247G, R174Q, or L155R) were detected in our patient samples, but they are not well-known hot spots of TP53. Collectively, these findings strongly suggest that YAP activation may serve as a clinical biomarker of resistance to ALK-TKI independent of genetic alterations.

## Discussion

In this study, we identified for the first time the activation of YAP as a potential therapeutic target for overcoming acquired resistance to ALK-TKI in *ALK*-rearranged NSCLC using cell lines, mouse models,

---

**Figure 4.  Effect of yes-associated protein (YAP) knockdown on cell growth and tumor growth *in vitro* and *in vivo*.**

A  Colony formation of the indicated cells treated with either dimethyl sulfoxide or crizotinib 24 h after siRNA transfection. Top: representative images for crystal violet staining. Bottom: quantification for crystal violet staining. **$P < 0.001$ vs. DMSO in Consi, #$P < 0.05$, ##$P < 0.01$ vs. the value at the indicated comparison in each cell lines. $n = 3$.

B  Top: representative immunoblots of total YAP levels in stably selected CR pool cells after infection with either negative control shRNA (Con sh) or YAP shRNA (YAP sh). Bottom: colony formation following treatment with either dimethyl sulfoxide or crizotinib in CR pool cells stably expressing Con sh or YAP sh.

C  Left: tumor growth curves of xenografts derived from CR pool cells stably expressing Con sh or YAP sh. **$P < 0.01$ vs. Con shRNA Middle: representative images showing tumorigenesis of Con sh (left flank) or YAP sh (right flank) stable CR pool cells implanted to nude mice. Right: comparison of the xenograft tumor volumes after 3 weeks from when tumor volumes reached 40–50 mm³ on the left. ***$P < 0.001$ vs. the value on day 0 in each group. ##$P < 0.01$ vs. the value for the indicated comparison. ns, not significant. $n = 6$.

D  Tumor growth curves of CR pool cell-derived xenografts during treatment with the indicated drugs. **$P < 0.01$ vs. vehicle. ##$P < 0.01$ vs. crizotinib treatment. ns, not significant. $n = 6$.

Data information: Data represent means ± SD (A, C and D). Kruskal–Wallis with Dunn's *post hoc* test was used for comparing multiple groups.

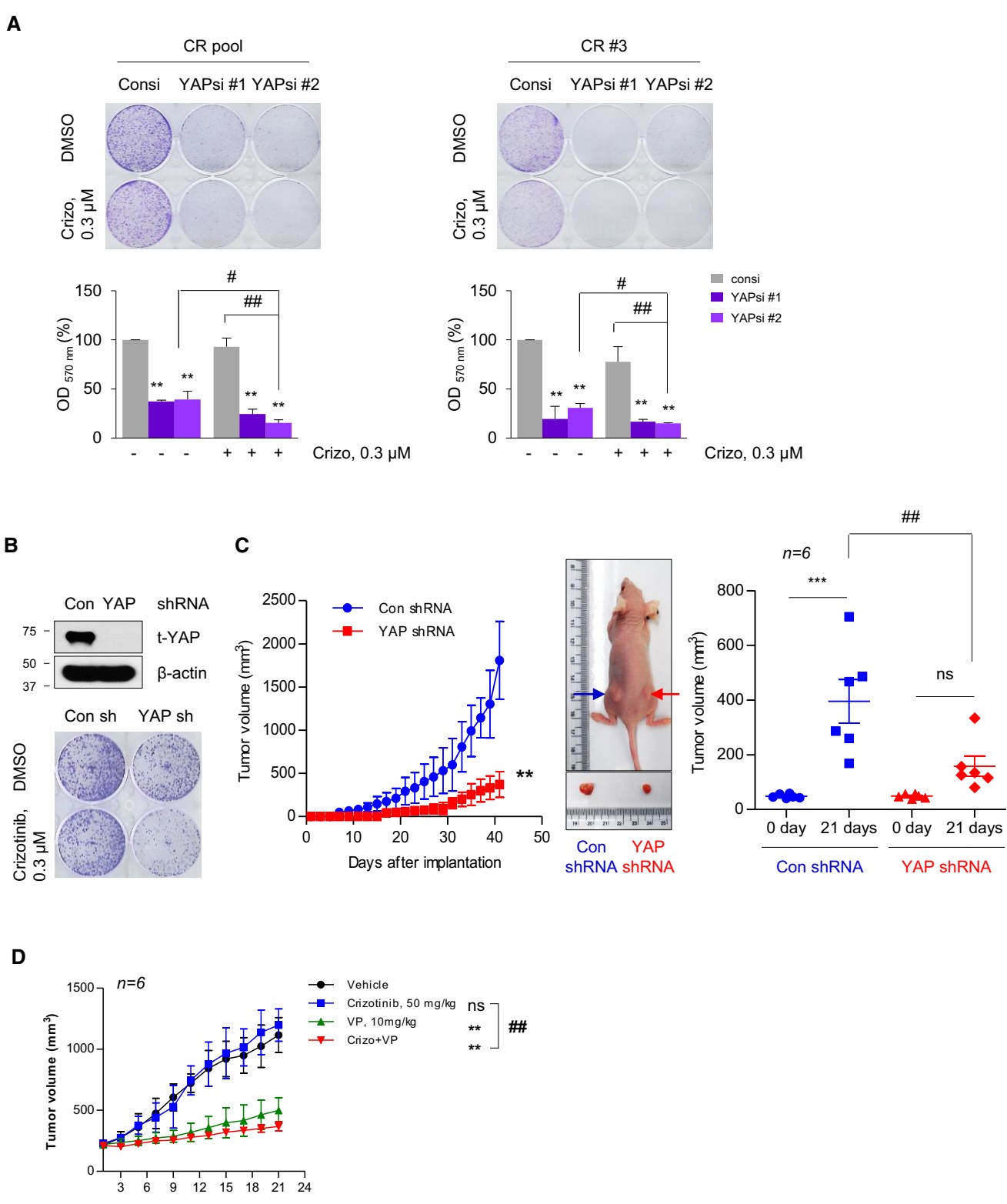

Figure 4.

and patient samples. Our findings also demonstrated that statins control transcriptional activity of YAP in geranylgeranylation-dependent manner, ultimately impacting the survival of resistant cells by modulating the expression of YAP-targeted oncogenic factors. This study proposes a preclinical rationale that targeting YAP may be a promising strategy for the treatment of ALK-TKI-resistant NSCLC.

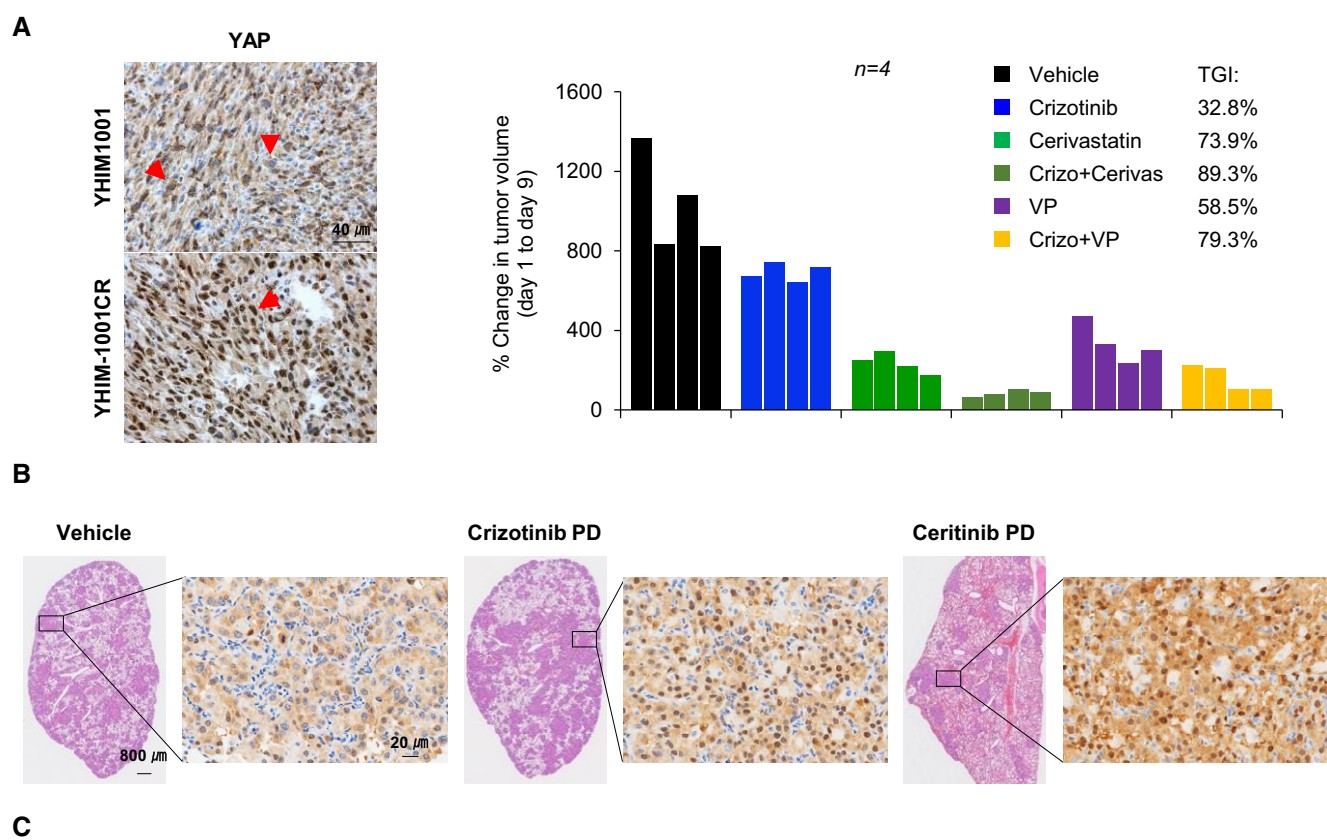

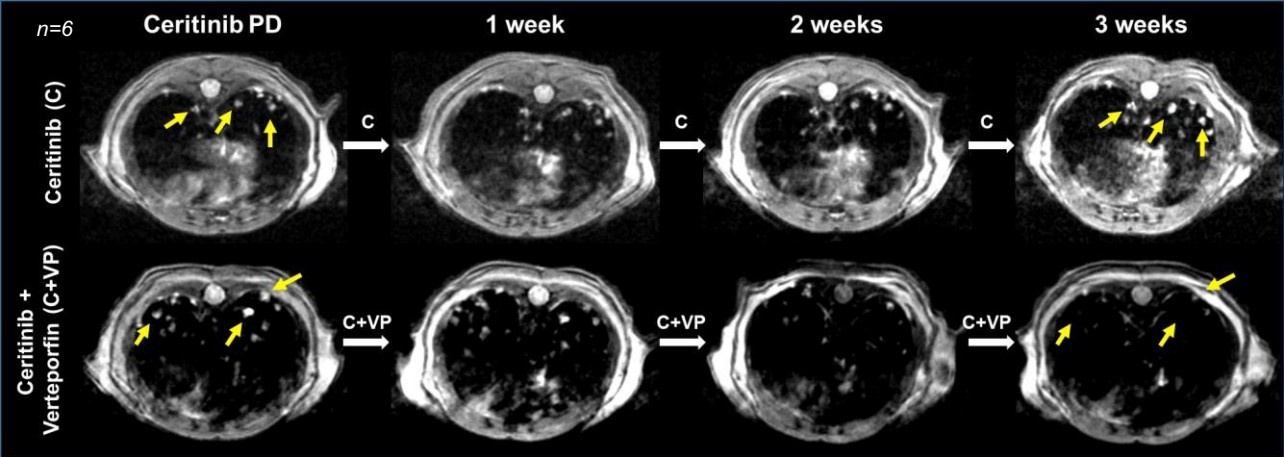

**Figure 5. Therapeutic activity of a combined treatment of crizotinib with either cerivastatin or VP in ALK-tyrosine kinase inhibitors-resistant ALK-positive patient-derived xenograft models and EML4-ALK transgenic mice.**

A   Left: representative IHC images of yes-associated protein (YAP) staining in tumor sections from YHIM1001 and YHIM1001CR. Right: individual tumor volumes of YHIM-1001CR xenografts treated with the indicated drugs. The red arrow indicates YAP staining.
B   Representative images of hematoxylin and eosin (H&E) and IHC staining of YAP in the indicated tumor sections from EML4-ALK transgenic mice.
C   Representative MRI images of two treatment groups (75 mg/kg ceritinib [C] and 75 mg/kg ceritinib + 10 mg/kg VP [C + VP]) with mice showing PD under ceritinib treatment. The yellow arrow indicates lung tumor nodules.

The mechanisms of acquired resistance to ALK-TKIs have been studied extensively using diverse experimental platforms and approaches. In particular, activation of EGFR (Katayama *et al*, 2012; Miyawaki *et al*, 2017), reactivation of the GTPase RAS–MAPK pathway (Crystal *et al*, 2014; Hrustanovic *et al*, 2015), and phenotypic changes such as EMT (Gainor *et al*, 2016; Wei *et al*, 2018) have been consistently reported in ALK-TKI-resistant models lacking secondary ALK alterations. Moreover, we and others have reported

Table 1.  Expression pattern of yes-associated protein (YAP) in pre- and post-ALK tyrosine kinase inhibitors paired samples (*n* = 17).

| Patients | Treatment | Duration of treatment (month) | Best response to ALK inhibitor | ALK secondary mutation (post-treatment) | YAPscore | | Nuclear YAP-positive | |
|---|---|---|---|---|---|---|---|---|
| | | | | | Pre | Post | Pre (%) | Post (%) |
| 1 | Ceritinib | 8 | SD | NA | 1 | NA | 0 | NA |
| 2 | Crizotinib | 13 | PR | NA | 2 | 2 | 50 | 80 |
| 3 | Crizotinib | 16 | PR | None | 2 | 3 | 20 | 80 |
| 4 | Ceritinib | 6 | SD | None | 3 | NA | 50 | NA |
| 5 | Ceritinib | 10 | PR | None | 2 | NA | 40 | NA |
| 6 | Crizotinib | 14 | PR | NA | 1 | NA | 20 | NA |
| 7 | Ceritinib | 2 | PD | None | 3 | NA | 0 | NA |
| 8 | Crizotinib | 22 | SD | None | 2 | 2 | 30 | 30 |
| 9 | Crizotinib | 10 | PR | L1196M | NA | 2 | NA | 60 |
| 10 | Crizotinib | 7 | PR | None | 1 | 2 | 0 | 10 |
| 11 | Crizotinib | 13 | SD | NA | 3 | 3 | 20 | 70 |
| 12 | Crizotinib | 17 | PR | NA | 3 | 3 | 10 | 50 |
| 13 | Crizotinib | 9 | PR | NA | 2 | 3 | 30 | 30 |
| 14 | Crizotinib | 13 | PR | NA | 2 | 3 | 50 | 80 |
| 15 | Crizotinib | 17 | PR | NA | 2 | NA | 10 | NA |
| 16 | Crizotinib | 27 | SD | NA | NA | 3 | NA | 80 |
| 17 | Crizotinib | 11 | PR | NA | 1 | 2 | 10 | 60 |

NA, not available tissue; PD, progressive disease; PR, partial response; SD, stable disease.

that AXL overexpression in an EMT phenotypic context is implicated as a mechanism of resistance to ALK-TKIs (Yun *et al*, 2018). Consistent with these reports, in present crizotinib-resistant *in vitro* models with cross-resistance to other ALK-TKIs, the expression of multiple targetable candidates such as EGFR, AXL, and CYR61 was concurrently increased, and expression levels of DUSP6 and IGFBP-3 were suppressed (Fig 2B–D and Appendix Fig S7A). Therefore, given these diverse resistance mechanisms, it is necessary to identify a master regulator of drug resistance to effectively overcome resistance to ALK-TKIs.

The Hippo signaling pathway is one of the 10 oncogenic signaling pathways with frequent genetic alterations, and Hippo pathway genes such as LATS1/2 and YAP were somatically mutated in 10% of 9,125 tumors across 33 cancers profiled by The Cancer Genome Atlas (Sanchez-Vega *et al*, 2018). YAP has recently been recognized as a key determinant that mediates drug resistance by acting in parallel to other signaling pathways (Lin *et al*, 2015; Gujral & Kirschner, 2017; Lee *et al*, 2018). Of note, overexpression of YAP and its target gene signatures has been reported to be associated with poor clinical outcomes in NSCLC patients (Wang *et al*, 2010). In line with previous reports, we demonstrated that a number of genes belonging to enriched pathways in crizotinib-resistant cells are functionally associated with YAP (Fig 2B and C and Appendix Fig S6). YAP silencing markedly diminished the expression of EGFR, AXL, and CYR61, suggesting a potential role of YAP in orchestrating multiple bypass signaling activation to ALK inhibition (Fig 2E). YAP knockdown also upregulated expression of the cyclin-dependent kinase inhibitor p21, resulting in a marked induction of cell cycle arrest at G0/G1 phase, thereby suppressing tumor cell proliferation of *in vitro/in vivo*-resistant

models (Figs 4 and EV4, and Appendix Fig S9). More importantly, YAP inhibition led to successful synergy in acquired-resistant ALK-positive PDX models and *EML4-ALK* transgenic mouse model against ALK-TKIs (Fig 5). Conversely, ectopic expression of YAP led to an incomplete response to crizotinib in ALK-positive *in vitro/in vivo* models accompanied with induction of CYR61, EGFR, and AXL (Figs 3 and EV3). Although some genes mentioned above have already been reported as YAP target genes (Zhao *et al*, 2008), the association between YAP transcriptional activity and resistance to ALK-TKI in *ALK*-rearranged NSCLC has not been reported. Our argument is strongly supported by our findings showing prominent nuclear YAP expression in ALK-TKI-resistant tumors compared with pretreatment tumors in *ALK*-rearranged NSCLC patients (Fig 6). We also observed a tendency toward high YAP expression in pretreatment specimens among non-responders (SD or PD) to ALK inhibitors than among responders, highlighting the capability of YAP as a survival input for ALK-rearranged NSCLC. A recent study on resistance to ALK inhibitors in multiple ALK inhibitor-resistant PDCs has revealed that inhibition of Src homology phosphotyrosine phosphatase 2 (SHP2) restores sensitivity to ALK inhibitors (Dardaei *et al*, 2018). Interestingly, several lines of evidence indicated that SHP2 interacts with the YAP/TAZ complex and that this interaction is essential for the oncogenic function of SHP2 (Kim *et al*, 2018). Thus, these previous findings potentiate our theory that YAP may serve as a central mediator of ALK-TKI resistance.

Growing evidence suggests that TAZ promotes resistance to various anti-cancer therapies including cytotoxic chemotherapy (Moroishi *et al*, 2015; Zhan *et al*, 2018; Liu *et al*, 2019). However, our results showed that TAZ knockdown had no inhibitory effect on

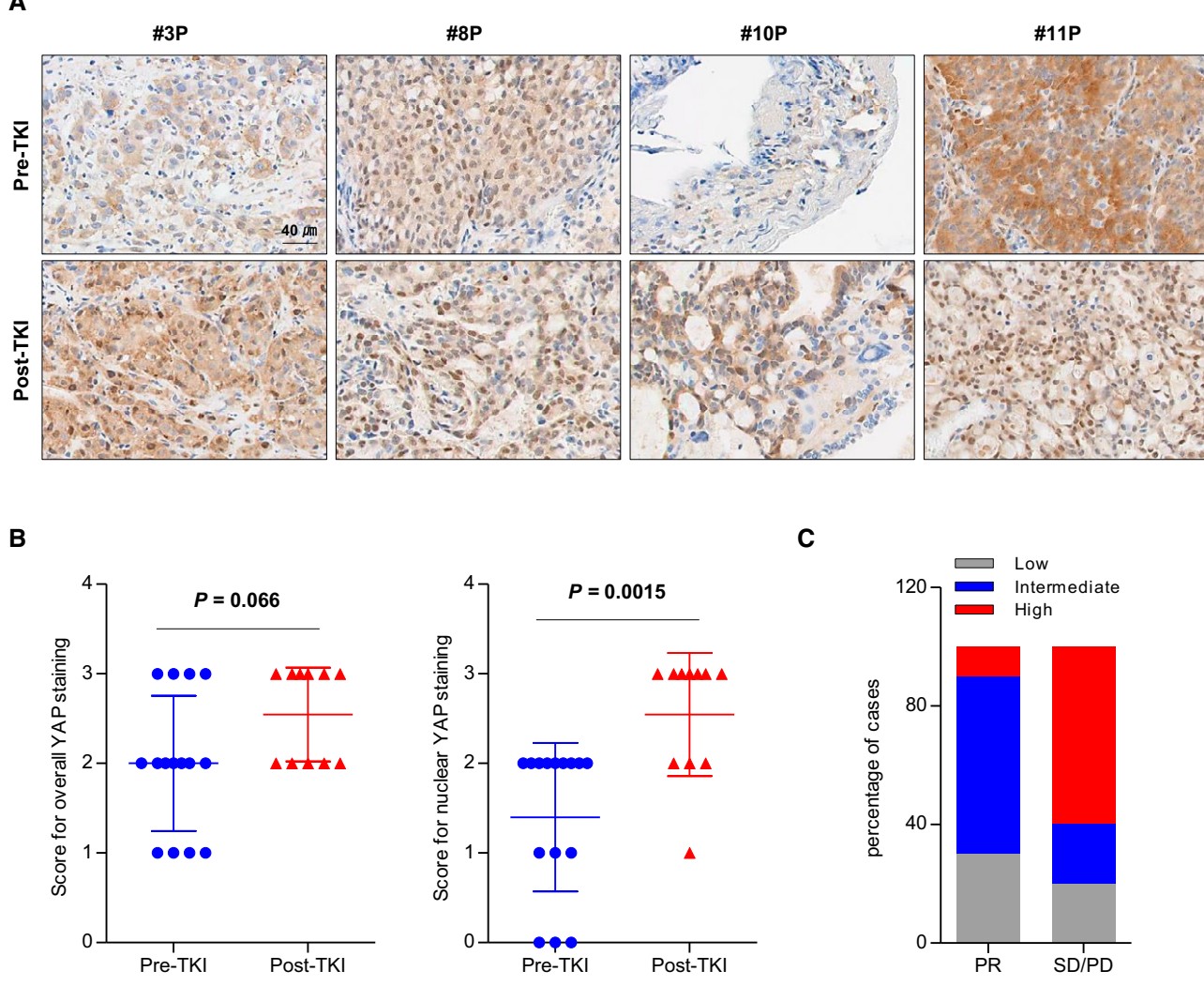

**Figure 6. IHC analysis of yes-associated protein (YAP) expression in pre- and post-treatment tumor biopsies from patients with ALK rearrangement.**

A Representative IHC images of YAP staining in pretreatment (pre-tyrosine kinase inhibitor [TKI], *n* = 15) and post-treatment (post-TKI, *n* = 11) tumor biopsies from patients treated with ALK-TKI (*n* = 26).

B Quantification of YAP IHC staining intensity (left) and YAP nuclear localization (right) based on IHC staining. Scoring was determined by calculating the sum of distribution scores and intensity scores as described in Materials and Methods. Each symbol represents individual tumor biopsies. Data represent means ± SD. *P*-values were determined using the Mann–Whitney test.

C Quantification of YAP expression according to the response criteria in pretreatment tumor biopsies. PR: partial response (*n* = 10), SD/PD: stable disease/progressive disease (*n* = 5).

the clonogenicity of CR and LR cells (Fig 4A and Appendix Figs S9 and S10). Interestingly, TAZ knockdown resulted in a marked increase in tumor suppressor IGFBP3 expression and complete inhibition of EMT-associated VIM expression, but YAP did not influence the expression of these genes (Appendix Fig S7). These results raise the possibility that TAZ may, at least in part, be associated with ALK-TKI resistance through distinct transcriptional events. Indeed, several studies have shown that YAP and TAZ regulate different downstream target genes by tissue-specific or cell type-specific transcription factor-binding partners (Varelas *et al*, 2008; Zhang *et al*, 2009). Moreover, Mi *et al* (2015) have reported that YAP mainly contributes to cell proliferation, while TAZ appears to regulate

migration in breast cancer. Therefore, it is necessary to further investigate the functional role of TAZ in ALK-TKI resistance.

It is well known that inhibition of the MVA pathway by statins impairs oncogenic functions of YAP/TAZ in a variety of cancer types (Sorrentino *et al*, 2014; Wang *et al*, 2014; Mi *et al*, 2015; Mullen *et al*, 2016). Here, we also showed anti-cancer activity of statins in resistant preclinical models by reducing nuclear localization and transcriptional activity of YAP (Figs 1 and EV1, and Appendix Fig S2). Notably, statin-mediated regulation of YAP target gene expression required GGPP, but was largely independent of MST/LATS kinase activity. In this regard, statins may represent effective therapeutic drugs against ALK-TKI resistance. In addition

to the preclinical anti-cancer effects of statin (Yang *et al*, 2017; Gordon *et al*, 2018), large-scale epidemiological studies have demonstrated the beneficial effects of long-standing statin use on lowering the risk of mortality of patients in diverse cancer types (Goss *et al*, 2016; Hung *et al*, 2017; Seckl *et al*, 2017). Although the clinical use of statins as anti-cancer agents may be limited due to undefined dosage/schedule and potential toxicity at micromolar concentrations (Holstein *et al*, 2006), the development of agents targeting the MVA pathway should be further explored considering the key role of MVA pathway metabolites in oncogenic activities of YAP.

In conclusion, MVA pathway-mediated YAP/TAZ activation converges to modulate key oncogenic processes, which are essential for driving resistance to ALK-TKI in ALK-rearranged NSCLC, and statin inhibits tumor growth and cell survival of ALK-TKI-resistant models by targeting YAP transcriptional activity. These findings indicate the need to develop YAP-targeting therapeutic agents to overcome acquired resistance to ALK-TKI in *ALK*-rearranged NSCLC.

# Materials and Methods

### Chemicals

Crizotinib, ceritinib, and lorlatinib were purchased from Selleck Chemical (Houston, TX). The library of FDA-approved drugs (Screen-Well FDA-Approved Drug Library, 640 chemical compounds dissolved at 10 mM in dimethyl sulfoxide) was obtained from Enzo Life Sciences (Plymouth Meeting, PA). The following compounds were purchased from Sigma Chemical Co. (St. Louis, MO): cerivastatin, simvastatin, fluvastatin, atorvastatin, GGPP, FPP, squalene, GGTI-298, FTI-277, and verteporfin. Drug preparation and use of all reagents were conducted according to manufacturer's instructions.

### Generation of ALK-TKI-resistant cells

H3122 cells were kindly provided by Dr Okamato from Kyushu University (Fukuoka, Japan). Cells were cultured in RPMI-1640 (Sigma-Aldrich) supplemented with 10% fetal bovine serum and 1% penicillin/streptomycin at 37°C in 5% $CO_2$, and the media were exchanged every 2–3 days. To generate crizotinib (CR)-/ceritinib (LDK, LR)-acquired-resistant models, we followed previously described protocols (Yun *et al*, 2018). Resistant cells (CR pool and LR pool; bulks of each drug-resistant cells, #1–10; independently derived resistant clones in colonies expanded from each pool) were derived after approximately 6 months of culture in the continuous presence of 1 μM crizotinib or ceritinib. CR pool, CR #1, CR #3, LR pool, and LR #6 resistant cells were used in all experiments.

### Cell viability assay

Cells were seeded onto 96-well plates ($1 \times 10^3$/well) and, after an overnight attachment period, were exposed to the various concentrations of appropriate drugs for 72 h. Cell viability was analyzed using Cell Titer Glo (Promega) according to manufacturer's protocol. The survival curves were calculated by Prism 5 software. For colony formation assays, siRNA-transfected single cells were seeded onto 6-well plates ($3 \times 10^3$/well) and, after an overnight attachment period, were exposed to the appropriate drugs for 7 days. Media including the corresponding concentration of drugs were replaced every 3 days. Upon treatment completion, cells were washed with phosphate-buffered saline (PBS), fixed in 4% paraformaldehyde for 10 min, and stained with 0.5% crystal violet for 20 min. To evaluate clonogenicity, images were captured using flatbed scanner and then the cells were dissolved with 20% SDS. The OD of the colony formation was read at 570 nm using a SpectraMax 250 microplate reader (Molecular Devices).

### Sanger sequencing

Genomic DNA was extracted from parental and resistant cells by the DNeasy Blood & Tissue Kit (Qiagen, Hilden, Germany) according to manufacturer's instructions. The entire ALK kinase domain was sequenced by Sanger dideoxynucleotide sequencing, and the entire coding region of the TP53 gene was sequenced using One-click Sanger Sequencing (Macrogen Inc, South Korea). Sequencing data were analyzed with Geneious v11.1.5 and NCBI BLAST. Using previously published reports (Rivlin *et al*, 2011; Freed-Pastor *et al*, 2012; Sorrentino *et al*, 2014; Parrales *et al*, 2016; Turrell *et al*, 2017), we listed 42 hot spots of TP53, including R270H mutation associated with the MVA pathway and then verified their presence in tested cell lines. Primer sequences are listed in Appendix Table S3.

### Immunofluorescence

Cells were washed with TBS, fixed with 4% paraformaldehyde for 30 min, permeabilized with 0.1% Triton X-100 for 10 min, and incubated with blocking solution (2% BSA in 0.1% Triton X-100 in TBS) for 30 min. The cells were then sequentially labeled with primary antibodies and the appropriate secondary antibodies diluted in blocking solution. The cells were mounted using prolong gold antifade reagent with DAPI and evaluated by a laser scanning confocal microscopy (LSM 510; Carl Zeiss, Jena).

### Immunoblot analysis

Cell lysates containing equal amounts of protein were separated by SDS–PAGE, transferred to membrane (Hybond; Amersham Biosciences, Piscataway, NJ), and immunoblotted with specific primary and secondary antibodies. The blots were detected by SuperSignal™ West Pico Chemiluminescent Substrate (Thermo Fisher Scientific, MA). The membrane was re-blotted with an anti-β-actin antibody as an internal control. Band intensities were quantified using an Alpha View SA (Cell Biosciences, Santa Clara, CA). For cell fractionation, the cells were centrifuged at 300 *g* for 5 min, and the pellets were suspended in hypotonic buffer for 15 min on ice. Lysates were centrifuged at 14,000 *g* for 30 s at 4°C, and the resultant supernatants (cytosolic fraction) were transferred to new tubes. The resultant pellets were suspended in extraction buffer and shaken at 4°C for 30 min on a shaking platform at 150 rpm. The nuclear extracts were centrifuged at 14,000 *g* for 10 min at 4°C, and the resultant supernatants (nuclear fractions) were frozen (−70°C). Lamin A/C was used as a nuclear marker and tubulin as a cytoplasmic marker.

**Flow cytometry**

To analyze cell cycle distribution, after appropriate drug treatment, both adherent and floating cells were harvested and fixed in cold 70% ethanol at 4°C for 4 h. The fixed cells were centrifuged at 300 *g* for 5 min, and the cell pellet was washed twice with ice-cold PBS and stained with 50 μg/ml propidium iodide (Sigma) solution containing 10 μg/ml RNase A. Cell cycle distribution was evaluated from 10,000 cells using a FACSVerse (BD Biosciences, CA, USA) running FlowJo V10.0.6 software.

**siRNA transfections**

Small interfering RNA transfections were performed with Lipofectamine™ RNAiMAX (Invitrogen, Carlsbad, CA) in antibiotic-free medium according to the manufacturer's instructions. Interference efficiency of the siRNA on target gene expression was evaluated by Western blot. Specific siRNAs for ALK (IDs: HSS105340 and HSS105342), YAP (IDs: HSS115944 and HSS115942), and Silencer™ Select Negative Control (#4390843) were obtained from Thermo Fisher Scientific (MA, USA). Specific siRNAs for TAZ (siGENOME WWTR1 #2, 4; MQ-016083-00-0002), LATS1 (E-004632-00-0020), LATS2 (E-003865-00-0020), and Accell Non-targeting pool (D-001910-10-20) were purchased from GE Dharmacon (Lafayette, CO).

**Microarray analysis**

Total RNA was hybridized to GeneChip® Human Gene 2.0 ST Arrays (Affymetrix, Santa Clara, CA) according to the manufacturer's instructions. Data analysis was performed using Affymetrix® Expression Console™ software. The resulting data are publically available via Gene Expression Omnibus accession GSE117181. Differentially expressed genes (DEGs) identified by using local pooled error test using R-based Bioconductor. Heat-maps were plotted using Morpheus (https://software.broadinstitute.org/morpheus/) and were generated based on a corrected *P*-value of < 0.05 and a fold change of > 1.5. Gene sets for YAP were obtained from previously published gene signature (Zhao *et al*, 2008; Zhang *et al*, 2009). Pathway analysis of DEGs with the *P*-value < 0.05 was performed in the Search Tool for the Retrieval of Interacting Genes (STRING; http://www.string-db.org) using the Kyoto Encyclopedia of Genes and Genomes (KEGG) pathway. *q*-value was controlled for multiple testing by determining the false discovery rate. A difference was considered significant if the *q*-value was < 0.05. To construct the network of genes belonging to TOP 15 KEGG pathway categories, including YAP gene, these genes were imported into the STRING and only interactions with a combined score of > 0.7 were pasted into the Cytoscape (http://www.cytoscape.org) plugin to create the network visualization.

**Plasmid construction and site-directed mutagenesis**

pBABE-YAP1 plasmid DNA (15682) was purchased from Addgene (Cambridge, MA). The YAP S127A mutant construct (serine to alanine at residue 127) was generated by the introduction of a single point mutation to pBABE-YAP1 using QuikChange Site-Directed PCR Mutagenesis Kit (Stratagene) and verified by sequencing. To exchange the backbone vector from retrovirus-based to lentivirus-based, insert PCR was performed with retroviral pBABE-YAP1 or YAP S127A plasmid as the template. TA cloning was subsequently performed using All in One™ PCR Cloning Kit (Biofact, Daejun, Korea) according to the manufacture's protocol. The multiple independent clones were isolated and confirmed by DNA sequencing. The correct plasmids were subcloned into lentiviral pLVX vector (Takara Bio Inc, Shimogyō-ku, Kyoto, Japan, cat #632164). The direction of the ligated fragment was confirmed by restriction mapping using EcoR I and Xba I enzymes.

**Establishment of YAP-overexpressing and YAP silenced stable cell lines**

For stable YAP-overexpressing cells, ectopic expression of empty control vector (pLVX), wild-type YAP (YAP-WT), or the YAP S127A mutant (YAP-S127A) in H3122 cell lines was achieved by a lentiviral-based approach. Transfection of lentiviral vectors together with lentiviral packaging mixtures into HEK293T cells (ATCC) was performed using lipofectamine 3000 reagent (Invitrogen, Carlsbad, CA) according to manufacturer's instructions. At 48 h post-transfection, the resulting lentiviral supernatant was collected and further filtered through a 0.45-μm pore filter and used to infect cells in the presence of 10 μg/ml polybrene. The transduced cells were then selected with 10 μg/ml puromycin to establish cells stably expressing YAP or YAP-S127A. For stable YAP knockdown cells, lentiviral particles for YAP shRNA (sc-38637-V; Santa Cruz Biotechnology, Dallas, TX) and control shRNA (sc-108080; Santa Cruz Biotechnology) were applied with 5 μg/ml polybrene to 60–70% confluent cells according to manufacturer's instructions.

**Animal studies**

All animal experiments were performed in compliance with worldwide standard animal care conditions by the Institutional Animal Care and Use Committee at Yonsei University College of Medicine. The research proposal was approved by the Association for Assessment and Accreditation of Laboratory Animal Care. Mice cages were limited to maximum of five animals per cage and checked daily for cage cleanliness and sufficient food/water.

We performed two types of *in vivo* studies using tumor xenograft models. To test the anti-tumor effect of statins in CR pool cell-derived xenografts, 6- to 8-week-old female nude mice (OrientBio, Seoul, Korea) were used and tumor xenograft models were generated by subcutaneous injection of $5 \times 10^6$ cells/0.1 ml PBS. Mice were randomly grouped (*n* = 6 mice per group) when tumors volume reached 150–200 mm³ and received the following treatments: vehicle, crizotinib (Crizo, 50 mg/kg, oral daily), cerivastatin (Cerivas, 1 mg/kg, intraperitoneal [i.p.] injection daily), atorvastatin (ATO, 10 mg/kg, i.p. daily) verteporfin (VP, 10 mg/kg, i.p. every 2 days) and combinations of Crizo with either Cerivas, ATO, or VP.

For *in vivo* tumorigenicity experiments, H3122 parental cell lines expressing YAP-WT or YAP-S127A ($5 \times 10^6$ cells/0.1 ml PBS for each cell line) were subcutaneously injected into the right flank of nude mice. pLVX cells with control vectors were injected into the left flank of the same mice. We then measured and documented tumor size every 3 days until tumors appeared, and monitored for a further 3 weeks after tumor volume reached 40–50 mm³. Similarly, CR pool cells expressing control shRNA or YAP shRNA

$(5 \times 10^6$ cells/0.1 ml PBS for each cell line) were subcutaneously injected into the left or right flank of nude mice, and tumor incidence was evaluated. Tumor volumes were measured twice a week to evaluate tumor growth rate, which was calculated using the formula: $0.532 \times \text{length} \times \text{width}^2$. % change in tumor volume was calculated according to the following formula: $(V_t - V_0)/V_0 \times 100$. TGI was calculated using the method of Drilon $et\ al$ (2018): $100\% \times [1 - (\text{TV}_t - \text{TV}_0)/(\text{CV}_t - \text{CV}_0)]$ where $V_0$ was the tumor volume at the beginning of the study, $V_t$ was the tumor volume at the end of the study, $\text{TV}_0$ was the $V_0$ in the treatment group, $\text{TV}_t$ was the $V_t$ in the treatment, $\text{CV}_0$ was the $V_0$ in the control group, and $\text{CV}_t$ was the $V_t$ in the control group.

### Immunohistochemistry

Following deparaffinization, immunohistochemistry (IHC) was performed following standard techniques. Briefly, sections were incubated in 3% $H_2O_2$ for 10 min after boiling in Tris–EDTA antigen retrieval buffers, blocked in 10% normal donkey serum, and incubated overnight at 4°C with primary antibody against human YAP antigens (Cell Signaling, Technology, Danvers, MA). The sections were washed and incubated with a biotinylated donkey anti-rabbit IgG (Jackson ImmunoResearch Laboratories, Inc., West Grove, PA) for 60 min and then visualized using the ImmunoCruz® ABC Staining System (Santa Cruz Biotechnology). The sections were lightly counterstained by hematoxylin and then dehydrated with alcohol and xylene The IHC evaluation was scored using following criteria: the YAP staining intensity (a scale from 0 to 3 was used, where 0 is negative, 1 is weak, 2 is moderate, and 3 is strong staining) and nuclear localization (the percentage of tumor cell nuclei-stained, where 0 is negative, 1 is < 10%, 2 is 10–50%, and 3 is > 50% of tumor cell nuclei-stained). All evaluations were independently reviewed by a pathologist (SHS) in a blinded manner. Antibodies are listed in Appendix Table S4.

### Generation of the crizotinib-resistant PDX model

$ALK$-positive PDX model (YHIM-1001) was established as previously described (Kang $et\ al$, 2018). For subcutaneous implantation of tumor tissues, 6-week-old female immunodeficient NOD/Shi-scid IL2rγ null (NOG) mice were used. We generated crizotinib-acquired-resistant PDX (YHIM-1001CR) by following a modified protocol of the technique described by Friboulet $et\ al$ (2014). Briefly, YHIM-1001 tumor-bearing nude mice were treated with increasing concentrations of crizotinib (50, 100, and 150 mg/kg) in a stepwise manner until tumors were no longer responsive to crizotinib. Tumors that continued to grow under 150 mg/kg crizotinib were isolated and transplanted into nude mice again followed by the treatment with vehicle or crizotinib. The mice were considered resistant to crizotinib if tumors continued to grow under 150 mg/kg crizotinib treatment.

### *In vivo* YAP studies in conditional EML4-ALK transgenic mice

We have previously produced conditional $EML4$-$ALK$ transgenic mice (Pyo $et\ al$, 2017; Yun $et\ al$, 2018). Briefly, 6-week-old male SPC-Cre-ERT2/EML4-ALK transgenic mice were treated with tamoxifen to induce EML4-ALK-mediated lung cancer. The mice were treated with either crizotinib (150 mg/kg orally) or ceritinib

(75 mg/kg orally), and magnetic resonance imaging (MRI) was performed to monitor the tumor response. Mice showed complete response to crizotinib or ceritinib within 2 weeks of treatment, but continued treatment led to disease progression.

### Tumor biopsy samples

All patient tumor biopsy samples were obtained from patients with ALK rearrangements before and after treatment with ALK-TKIs at the Yonsei University Severance Hospital (Republic of Korea), the National University Cancer Institute (Singapore), and the National Taiwan University Hospital (Taiwan). The study protocol was approved by the Institutional Review Board of each Institute. Informed consent was obtained for all patients, and experiments were conformed to the principles set out in the World Medical Association Declaration of Helsinki and the United States Department of Health and Human Services Belmont Report.

### Sequencing of patient samples

All available post-treatment biopsy samples were analyzed for ALK mutations conferring drug resistance. Testing was performed using the Ion Ampliseq™ hotspot cancer panel, which includes 50 cancer-related genes, including ALK. Tumor genomic DNA was extracted from formalin-fixed, paraffin-embedded (FFPE) tumor tissue blocks, and the library was prepared. The pooled capture library was quantified using Qubit (Invitrogen) and Tape Station (Agilent Technologies, Santa Clara, CA) and sequenced with an Ion Proton™ System. Sequencing data and ALK-resistant mutations were processed and reported as previously described (Gainor $et\ al$, 2016).

### Statistical analysis

Statistical analysis was performed using Mann–Whitney test for comparisons of two groups or by Kruskal–Wallis with Dunn's $post\ hoc$ test or ANOVA with Tukey's $post\ hoc$ test for comparisons of multiple groups, appropriately. Computations were performed using SPSS and GraphPad Prism software. Cell line experiments were independently repeated more than three times, with technical triplicate in each condition. For animal studies, the mice were randomly grouped when a tumor reached 200 $\text{mm}^3$ size before drug treatment. Tumor growth was measured by two individuals independently. The staining intensity for IHC studies was blindly assessed by a pathologist. Data in figures are expressed as the means $\pm$ SD or $\pm$ SE for three or more individual experiments. $P < 0.05$ was considered significant. Exact $P$-values are included in Appendix Table S5. All statistical analyses were all reviewed by our statistical collaborators.

**Expanded View** for this article is available online.

### Acknowledgements

We thank all the patients who donated samples for this study. This research was supported by Basic Science Research Program through the National Research Foundation of Korea (NRF) funded by the Ministry of Science, ICT & Future Planning (2016R1A2B3016282 to BCC) and the NRF grant funded by the Korean government (2014R1A1A1006865 to MRY). The authors thank MID (Medical Illustration & Design) for helping to design the figures.

## The paper explained

### Problem

Patients with non-small cell lung cancer (NSCLC) harboring the echinoderm microtubule-associated protein-like 4 (EML4)–ALK fusion exhibit remarkable initial responses to ALK-TKIs. However, durable clinical benefit has been limited by the development of acquired resistance. Several mechanisms of acquired resistance to ALK inhibitors have been proposed, but major mechanisms that explain widespread resistance emergence remain unknown.

### Results

To investigate a possible candidate for overcoming acquired resistance to ALK-TKI, we established acquired resistance models to ALK-TKIs (H3122-CR and H3122-LR) by continually exposing *EML4-ALK*-positive H3122 lung cancer cells to increasing doses of crizotinib or ceritinib, after which we performed cell viability screening using a library of FDA-approved drugs composed of a collection of 640 clinically used compounds. For the first time, we identified YAP activation as a master regulator of multiple resistance factors that potentially confer resistance to ALK inhibitors. YAP inhibition suppressed tumor growth in resistant cells, patient-derived xenograft, and *EML4-ALK* transgenic mice, whereas YAP overexpression decreased the responsiveness of parental cells to ALK inhibitor. We further verified the clinical relevance of active YAP in tumor biopsies of patients with resistance to ALK inhibitors.

### Impact

Our findings highlight GGPP-mediated YAP activation as a novel mechanism of crizotinib resistance and provide a rationale that targeting YAP may be a potential treatment strategy for *EML4-ALK*-positive NSCLC patients who have acquired resistance to ALK-TKIs.

## Author contributions

MRY designed experiments, analyzed the data, generated the figures, and wrote the manuscript. HMC performed most of the experiments. YWL and JWC analyzed the microarray experiments. HSJ and CWP conducted site-directed mutagenesis and generated the stable cell lines. CWP and K-HP generated conditional EML4-ALK transgenic mice. HNK and EJS generated *ALK*-positive PDX model (YHIM-1001). HSS evaluated scoring for IHC experiments. RAS, JC-HY, HC, MHH, and HRK collected the human lung cancer tissue specimens and contributed to their analysis using IHC. DHK, SSL, MHK MHH, and HRK edited the manuscript. BCC conceived this study, supervised it, and wrote and edited the manuscript.

## Conflict of interest

The authors declare that they have no conflict of interest.

## For more information

(i)   http://je-uklab.yuhs.ac/

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
