## [Review Process File · EMBO Molecular Medicine]

Targeting YAP to overcome acquired resistance to ALK inhibitors in ALK-rearranged lung cancer

Mi Ran Yun, Hun Mi Choi, You Won Lee, Hyeong Seok Joo, Chae Won Park, Jae Woo Choi, Dong Hwi Kim, Han Na Kang, Kyoung-Ho Pyo, Eun Joo Shin, Hyo Sup Shim, Ross A Soo, James Chih-Hsin Yang, Sung Sook Lee, Hyun Chang, Min Hwan Kim, Min Hee Hong, Hye Ryun Kim, Byoung Chul Cho

Review timeline:

Submission date:	11 March 2019
Editorial Decision:	3 May 2019
Revision received:	31 July 2019
Editorial Decision:	4 September 2019
Revision received:	20 September 2019
Accepted:	24 September 2019

Editor: Céline Carret

Transaction Report:

1st Editorial Decision

3 May 2019

Thank you for the submission of your manuscript to EMBO Molecular Medicine. We have now heard back from the three referees whom we asked to evaluate your manuscript.

You will see that the three reports are consistent and overall supportive. Ref. #1 requests testing another type of statin due to reported cerivastatin fatal side-effects and investigate p53 mutant role in the reported process, which should be investigated along with performing some missing controls and amending literature. Ref. #2 recommends silencing TAZ to improve conclusiveness and providing missing information/explanation. Further, upon our cross-commenting exercise, this referee agreed on the necessity to discuss the translational challenges posed by cerivastatin side effects and it would be informative (and straightforward) to verify p53 status in resistant line(s).

We would therefore welcome the submission of a revised version within three months for further consideration and would like to encourage you to address all the criticisms raised as suggested to improve conclusiveness and clarity. Please note that EMBO Molecular Medicine strongly supports a single round of revision and that, as acceptance or rejection of the manuscript will depend on another round of review, your responses should be as complete as possible.

I look forward to receiving your revised manuscript.

***** Reviewer's comments *****

Referee #1 (Remarks for Author):

The authors study mechanisms of resistance to clinically available targeted therapy (crizotinib) in EML4-ALK fusion positive non small cell lung cancer (NSCLC) preclinical models and identify resistant variants that do not have mutations in ALK. They go on to screen a library of 640 FDA approved drugs and find that the statin inhibitor cerivastatin had anti-cancer activity in crizotinib resistant ALK+ tumors. They also found upregulation of YAP function in these resistant tumors and go on to study the effect of genetically and pharmacologically inhibiting YAP function. Cerivastatin downregulated YAP function and combined treatment with crizotinib and cerivastatin led to marked anti-tumor effects in xenograft models including patient derived xenografts, and genetically engineered mouse models (GEMMs) of ALK+ NSCLC. In study of patient ALK+ NSCLC tumor samples before and after targeted therapy they found increased YAP function predicted less response if present initially and that in tumors that were initially sensitive to targeted therapy, YAP function was increased after targeted therapy resistance developed. They conclude: " Our findings highlight a crucial role of YAP in ALK-TKI resistance and provide a rationale for targeting YAP as a potential treatment option for ALK-rearranged patients with acquired-resistance to ALK inhibitors. "

Comments to the Authors:

This paper is reviewed in the context of the need to better understand and provide new therapeutic targets for ALK+ NSCLC and to overcome resistance to available targeted therapy, and the evolving knowledge of YAP in tumor formation and resistance to therapy. Overall the authors have provided a compelling and interesting story that has significant clinical translational application for ALK+ NSCLC. An important part of their findings is that combined therapy with ALK+ targeted therapy and cerivastatin targeted therapy has large quantitative anti-tumor effects in xenograft models. However, there are several issues the authors need to address.

1. The most important implication for clinical translation is the availability of an "FDA approved" drug in cerivastatin a statin anti-cholesterol drug which in the past was in widespread use clinically. However, cerivastatin was removed from the market in 2002 because of 52 deaths related to rhabdomyolysis and this is never mentioned by the authors. As important, the obvious next question is what about the effects of statins that are currently approved. We need to know if one of them can work as well, particularly in a xenograft. The answer maybe "no" and that is alright - and then that means the work needs to be targeted toward working on a cerivastatin derivative and exploring what the differences are between cerivastatin and the other statins to lead to the presence or absence of anti-tumor responses.
2. The use of verteporfin as a pharmacologic YAP inhibitor is clearly important. However, the authors do not cite any of the multiple papers that demonstrate it is a YAP inhibitor and papers that show it may have other effects. Also, verteporfin is in clinical use as a photosensitizer. While there are reports that it can inhibit YAP without light, an obvious question is whether light and photosensitization played a role in the anti-tumor effects seen. There is no discussion of this, and no mention in the methods or figure legends whether the experiments were controlled for light exposure. In at least some experiments this would be crucial to do. Also, the spelling of verteporfin should be the same throughout.
3. There are reports where mutant p53 interacts with the mevalonate pathway to lead to mutant p53 stabilization and upregulated YAP/TAZ function. The authors need to provide information on the p53 mutation status of the tumor lines and xenografts they tested and determine if mutant p53 plays an important role in the process they are describing. (Parrales A, Thoenen E, Iwakuma T. The interplay between mutant p53 and the mevalonate pathway. *Cell Death Differ.* 2018;25(3):460-70. PMC5864191.)
4. There are multiple examples of where English grammar needs to be corrected.

Referee #2 (Remarks for Author):

In the manuscript "Targeting YAP to overcome acquired resistance to ALK inhibitors in ALK-rearranged lung cancer" Yun et al. investigate mechanisms leading to resistance to ALK inhibitors in NSCLC lines and PDXs. The Authors discover that inhibition of the Mevalonate pathway sensitizes

resistant cells to ALK inhibitors and link this to the activation of YAP. Loss of function and rescue experiments support a role of YAP in mediating resistance to ALK-inhibitors. Although YAP has already been linked to therapy resistance, its role in ALK-targeted therapy has not been addressed, thus the present work is novel and relevant. Overall the study is well designed and executed, there are only few points that need to be clarified.

Major points

1. Figure 1E: is there a reason why FPP fails to rescue viability when used in combination with Cerivastatin? This is rather unexpected since GGPP (which is downstream FPP) is instead rescuing.
2. Regarding the potential role of TAZ in mediating chemosensitivity/resistance, based on the transcriptional effect of TAZ knockdown, the Authors conclude that YAP (but not TAZ) is the target of Cerivastatin. The fact that loss of TAZ does not recapitulate all the transcriptional changes observed when treating with Cerivastatin does not imply that TAZ is not relevant. Authors need to silence TAZ and assess whether this will sensitize resistant cells (CR and LR cells) to ALK inhibitors.
3. Page 10: "However, YAP suppression could not restore the sensitivity to crizotinib due to ALK-independent resistance mechanism in resistant cells (Fig. 4D)." . This observation is unexpected and at odds with the Authors main claim. Did the Authors check the level of YAP knockdown at the end of the experiment? This unexpected result could be due to clonal selection of cells with high YAP levels.

Minor points

1. Why is Cerivastatin treatment leading to p21 increase, is p21 repressed by YAP? Is p53 activated?
2. Figure 2B: Why is YAP upregulated when cells are treated with GGPP and cerivastatin?
3. Figure 2C,D: What are the values shown in the heatmaps?
4. Figures 1, 3 and 4: please indicate the number of mice used in the experiments shown.
5. Page 11: "In xenograft models, following subcutaneous cell injection, tumor from control cell were mostly observed 2 weeks, but those from stable YAP- knockdown cells were started to appear about 1 month and consequently formed smaller tumors at the end of an experiment (Fig. 4B and 4C)". Authors need to rephrase this: "within 2 weeks", "appear in about"
6. Page 12: "Of note, 60% (3/5) of tumors showing stable disease (SD) or progressive disease (PD) as the best response to ALK-TKI had high staining intensity, whereas 90% (9/10) of partial response (PR) samples showed low to intermediate staining intensity in pre-treatment tumor biopsies (Fig. 6C)". Please specify if here Authors are referring to YAP staining.

Referee #3 (Remarks for Author):

The manuscript by Yun and coworkers investigates the role of YAP signaling to treat NSCLC that have become resistant to ALK inhibitor therapy. This group has observed that some NSCLC which are treated with ALK inhibitor therapy. become resistant to therapy. While some have ALK dependent resistance, this group try's to find the cause and treatment for the cancers that become resistant independent of ALK signaling. using a ALK resistant cell line generated by this group, the group performed a screen of a 640 FDA approved drug library. They found that cerivastatin which targets the mevalonate pathway was the most selective. They then demonstrate that cerivastatin targets the YAP pathway and go to evaluate the role of the YAP pathway in the regulation of lng cancer cell growth in vitro and in patient derived xenograph models. They also demonstrate that nuclear localization of YAP is a clinically relevant alteration in human lung cancer. All in all this is a thorough and well designed manuscript. They conducted a screen for a new therapy to treat ALK inhibitor resistant lung cancer and have identified a novel pathway to target customized therapy for the treatment of lung cancer. All in all this is a novel and important study.

Referee #1:

This paper is reviewed in the context of the need to better understand and provide new therapeutic targets for ALK+ NSCLC and to overcome resistance to available targeted therapy, and the evolving

knowledge of YAP in tumor formation and resistance to therapy. Overall the authors have provided a compelling and interesting story that has significant clinical translational application for ALK+ NSCLC. An important part of their findings is that combined therapy with ALK+ targeted therapy and cerivastatin targeted therapy has large quantitative anti-tumor effects in xenograft models. However, there are several issues the authors need to address.

Response: We thank the Reviewers for their insightful and positive comments and suggestions. Our specific responses to the reviewers' comments are individually detailed below. We hope that this revised text makes our manuscript more suitable for publication in *EMBO Molecular Medicine*. Once again, we really appreciate that you have kindly edited and improved our manuscript. Thank you for your comments and courtesy.

1. The most important implication for clinical translation is the availability of an "FDA approved" drug in cerivastatin a statin anti-cholesterol drug which in the past was in widespread use clinically. However, cerivastatin was removed from the market in 2002 because of 52 deaths related to rhabdomyolysis and this is never mentioned by the authors. As important, the obvious next question is what about the effects of statins that are currently approved. We need to know if one of them can work as well, particularly in a xenograft. The answer maybe "no" and that is alright - and then that means the work needs to be targeted toward working on a cerivastatin derivative and exploring what the differences are between cerivastatin and the other statins to lead to the presence or absence of anti-tumor responses.

Response: Thank you for your insightful comments. In our original supplementary Figure 2, we showed the effect of simvastatin, another clinically available statin, on the survival of CR cells. Although the IC_{50} value of simvastatin was 10 times higher than that of cerivastatin, it was within acceptable range in other preclinical studies. Moreover, similar to cerivastatin, simvastatin successfully induced c-PARP, c-Cas3 and p21 expression in CR cells (These results are represented in Appendix Fig S2 of the revised Figure).

Appendix Figure S2. *In vitro* anti-proliferative effect of simvastatin.

A Cell viability curve in response to combined treatment of simvastatin and crizotinib in parental and CR cells using MTT assays.

B Representative immunoblots of the indicated proteins in lysates of cells treated with either cerivastatin (Ceriva) or simvastatin (Simva) for 24 h ($n = 3$). Blots are representative of three independent experiments.

Since we assumed that differences in IC_{50} values between the statins were due to their pharmacological properties (e.g., rate of absorption and solubility), we used cerivastatin as a representative in subsequent experiments.

However, considering that cerivastatin was withdrawn from the market, due to fatal rhabdomyolysis and kidney failure (Furberg & Pitt, 2001), there is insufficient evidence to claim the possibility of using statins as ALK-TKI-resistant therapies in our study.

Thus, we performed additional *in vitro* and *in vivo* experiments on atorvastatin in CR cells. As shown in Figure EV1 below, treatment with atorvastatin also showed similar results to the *in vitro* and *in vivo* anti-cancer activity of cerivastatin and simvastatin against CR cells.

We also included additional information in the Results (at 2nd paragraph of page 6) of the revised manuscript as follows. “Considering that cerivastatin was withdrawn from the market in 2001 due to fatal rhabdomyolysis and kidney failure (Furberg & Pitt, 2001), we additionally evaluated the anti-proliferative capability of other the clinically available statins atorvastatin and simvastatin in CR cells. Although they were used at higher concentrations than cerivastatin, these statins successfully inhibited cell growth at the concentration range reported in other previous preclinical studies (Fig EV1A and Appendix Fig S2A). Both cerivastatin, simvastatin and atorvastatin remarkably increased the expression levels of c-PARP, c-Cas3, and p21 in CR cells (Fig EV1B, and Appendix Fig S2B). These *in vitro* findings were further confirmed by *in vivo* xenograft studies showing that cerivastatin and atorvastatin significantly delayed tumor growth of the CR pool (Fig 1C and Fig EV1C). Based on the anti-cancer effects of statins, cerivastatin with the lowest IC₅₀ was used as a representative in subsequent experiments despite being a clinically discontinued drug.”

Figure EV1. *In vitro* and *in vivo* anti-cancer activity of atorvastatin.

A Cell viability curve in response to combined treatment of simvastatin and crizotinib in parental and CR cells using MTT assays. Data represent means \pm SD

B Representative immunoblots of the indicated proteins in lysates of cells treated with atorvastatin (ATO) for 24 h.

C Tumor growth curves of CR pool xenografts (n = 6) treated with the indicated drugs (**P* < 0.05 vs. vehicle, #*P* < 0.05 vs. crizotinib treatment. ns, not significant. Kruskal-Wallis followed by Dunn's post hoc test).

D, E Representative immunoblots of the indicated proteins in cells treated with ATO (5 μM) alone or with GGPP (10 μM) for 24 h.

Blots are representative of three independent experiments.

Reference

1. Furberg CD, Pitt B (2001) Withdrawal of cerivastatin from the world market. *Curr Control Trials Cardiovasc Med.* 2(5): 205–207

2-1. The use of verteporfin as a pharmacologic YAP inhibitor is clearly important. However, the authors do not cite any of the multiple papers that demonstrate it is a YAP inhibitor and papers that show it may have other effects.

Response: We apologize for missing references to the mentioned sentences. So, we revised as following as: “Verteporfin (VP), a pharmacological inhibitor of YAP, is known to inhibit YAP transcriptional activity by preventing the interaction of YAP and TEA domain family members (TEAD) (Liu-Chittenden et al., 2012; Yu FX et al., 2014).

This sentence was inserted at the 1st paragraph of the page 8 in the Results of the revised manuscript.

References

1. Liu-Chittenden Y, Huang B, Shim JS, Chen Q, Lee SJ, Anders RA, Liu JO, Pan D (2012) Genetic and pharmacological disruption of the TEAD-YAP complex suppresses the oncogenic activity of YAP. *Genes Dev* 26: 1300–1305

2. Yu FX, Luo J, Mo JS, Liu G, Kim YC, Meng Z, Zhao L, Peyman G, Ouyang H, Jiang W et al (2014) Mutant Gq/11 promote uveal melanoma tumorigenesis by activating YAP. *Cancer Cell* 25: 822–830

2-2. Also, verteporfin is in clinical use as a photosensitizer. While there are reports that it can inhibit YAP without light, an obvious question is whether light and photosensitization played a role in the anti-tumor effects seen. There is no discussion of this, and no mention in the methods or figure legends whether the experiments were controlled for light exposure. In at least some experiments this would be crucial to do.

Response: We sincerely acknowledge your comments. We have used verteporfin while protecting it from light, according to manufacturer's instructions. Thus, we believe that the anti-cancer activity of verteporfin in the ALK-TKI resistant model is independent of light activation.

Moreover, we have added the following sentence “Drug preparation and use of all reagents was conducted according to manufacturer's instructions.” in the “Chemicals” subsection in Materials and Methods of the revised manuscript.

We have also clarified in the figure legends of all experiments using verteporfin that the drug was used under protection from light at all steps.

Finally, we clarified this in the Result (at the 2nd paragraph of the page 9) as follows:

“Considering that VP has been clinically used as a photosensitizer in photodynamic therapy (Bressler *et al*, 2000; Battaglia *et al*, 2016), our results showed that VP exhibits significant therapeutic effects against ALK-TKI resistance by inhibiting YAP transcriptional activity without light activation, which is consistent with other reports (Brodowska *et al*, 2014; Slemmons *et al*, 2015; Ma *et al*, 2016; Cheng *et al*, 2016).”

References

1. Bressler NM, Bressler SB (2000) Photodynamic therapy with verteporfin (Visudyne): impact on ophthalmology and visual sciences. *Invest Ophthalmol Vis Sci* 41: 624–628

2. Battaglia Parodi M, La Spina C, Berchicci L, Petrucci G and Bandello F (2016) Photosensitizers and Photodynamic Therapy: Verteporfin. *Dev Ophthalmol* 55: 330–336

3. Brodowska K, Al-Moujahed A, Marmalidou A, Meyer Zu Horste M, Cichy J, Miller JW, Gragoudas E, Vavvas DG (2014) The clinically used photosensitizer Verteporfin (VP) inhibits YAP-TEAD and human retinoblastoma cell growth in vitro without light activation. *Exp Eye Res* 124: 67–73

4. Slemmons KK, Crose LE, Rudzinski E, Bentley RC and Linardic CM (2015) Role of the YAP Oncoprotein in Priming Ras-Driven Rhabdomyosarcoma. *PLoS One* 10: e0140781

5. Ma YW, Liu YZ, Pan JX. (2016) Verteporfin induces apoptosis and eliminates cancer stem-like cells in uveal melanoma in the absence of light activation. *Am J Cancer Res* 1: 2816–2830.
6. Cheng H, Zhang Z, Rodriguez-Barrueco R, Borczuk A, Liu H, Yu J, Silva JM, Cheng SK, Perez-Soler R, Halmos B (2016) Functional genomics screen identifies YAP1 as a key determinant to enhance treatment sensitivity in lung cancer cells. *Oncotarget* 7: 28976–28988

2-3. Also, the spelling of verteporfin should be the same throughout.

Response: We apologize for the oversight and have maintained consistency in spelling.

3. There are reports where mutant p53 interacts with the mevalonate pathway to lead to mutant p53 stabilization and upregulated YAP/TAZ function. The authors need to provide information on the p53 mutation status of the tumor lines and xenografts they tested and determine if mutant p53 plays an important role in the process they are describing. (Parrales A, Thoenen E, Iwakuma T. The interplay between mutant p53 and the mevalonate pathway. *Cell Death Differ.* 2018;25(3):460-70. PMC5864191.)

Response: We absolutely agree with your comments that we should provide information regarding the biological link between mutant p53 and the mechanism underlying YAP activation in the ALK-resistant model. Accordingly, we additionally investigated p53 mutational status in our tested cell line models by Sanger sequencing. Using previously published reports (Rivlin *et al.*, 2011; Freed-Pastor *et al.*, 2012; Sorrentino *et al.*, 2014; Parrales *et al.*, 2016; Turrell *et al.*, 2017), we listed 42 hot spots of TP53, including R270H mutation associated with the mevalonate pathway, and then verified their presence in tested cell lines. As shown in Appendix Table S1 below, only the E285V mutant was detected in parental cells, which is consistent with the cell line gene mutation profile of H3122 in COSMIC (<http://www.sanger.ac.uk/cosmic/>). Resistant cells did not exhibit any mutations other than E285V.

We further included the following sentence in Results (at lines 6 of page 7).

“Reportedly, the MVA pathway is associated with mutant p53 expression in a variety of cancer types (Freed-Pastor *et al.*, 2012; Sorrentino *et al.*, 2014; Parrales A *et al.*, 2016; Turrell *et al.*, 2017). However, there was no difference in the p53 mutational status between H3122 parental and resistant cells (Appendix Table S1). These results suggest that anti-cancer activity of statin in resistant cells may be independent of TP53 mutation.”

Variant	Alteration	Parental	CR pool	CR #1	CR #3	Variant	Alteration	Parental	CR pool	CR #1	CR #3
A159D	c.476C>A					R175L	c.524G>T				
A159P	c.475G>C					R248G	c.742C>G				
A159V	c.476C>T					R248L	c.743G>T				
E285V	c.854A>T	detected	detected	detected	detected	R248P	c.743G>C				
G245A	c.734G>C					R248Q	c.743G>A				
G245C	733G>T					R248W	c.742C>T				
G245D	c.734G>A					R249M	c.746G>T				
G245F	c.733_734GG>TT					R249S	c.747G>T				
G245N	c.733_734GG>AA					R273C	c.817C>T				
G245R	c.733G>C					R273G	c.817C>G				
G245S	c.733G>A					R273H	c.818G>A				
G245V	c.734G>T					R273L	c.818G>T				
R158C	c.472C>T					R273P	c.818G>C				
R158G	c.472C>G					R273S	c.817C>A				
R158H	c.473G>A					R280K	c.839G>A				
R158L	c.473G>T					R282W	c.844C>T				
R158P	c.473G>C					V157D	c.470T>A				
R158S	c.472C>A					V157F	c.469G>T				
R175C	c.523C>T					V157G	c.470T>G				
R175G	c.523C>G					V157L	c.469G>C				
R175H	c.524G>A					Y220C	c.659A>G				

Appendix Table S1. TP53 mutational status in crizotinib-resistant cells compared with H3122 parental cells

We also confirmed that silencing of YAP/TAZ as well as treatment with cerivastatin or GGTI-298 did not affect p53 expression in CR cells, indicating that the MVA pathway-YAP axis in resistant cells may be independent of TP53 mutations.

Additional Figure. Effect of MVA-YAP axis in p53 expression

A Representative immunoblots of p53 in CR cells after treatment of cerivastatin or GGTI-298 for 24 h.

B Representative immunoblots of p53 in lysates of CR cells transfected with either negative control siRNA (Con si), YAP siRNAs (two sets of siRNAs against YAP; YAP si#1 and YAP si#2), TAZ siRNAs (two sets of siRNAs against TAZ; TAZ si#1 and TAZ si#2), or a combination of TAZ siRNAs with YAP si #2 (Y+T si #1 and Y+T si #2).

However, these findings are not included in the revised manuscript due to space limitations. We hope you would understand our circumstances.

Furthermore, in our original Supplementary Table 2, we showed the genetic mutation status of the 7 available post-ALK TKI biopsy samples by targeted sequencing. Several TP53 mutations (P33R, E247G, R174Q or L155R) were detected in biopsy samples, but they are not well-known TP53 hot spots of TP53 and are not involved in statin sensitivity (These results are presented in Table EV2 and Appendix Table S2 of the revised Figure).

We also included the following sentence in the Results of the revised manuscript.

“Moreover, several TP53 mutations (P33R, E247G, R174Q or L155R) were detected in our patient samples, but they are not well-known hot spots of TP53.”

And this sentence “The entire coding region of the TP53 gene was sequenced using One-click Sanger Sequencing (Macrogen Inc, South Korea). Sequencing data were analyzed with Geneious v11.1.5 and NCBI BLAST. Using previously published reports (Rivlin *et al*, 2011; Freed-Pastor *et al*, 2012; Sorrentino *et al*, 2014; Parrales *et al*, 2016; Turrell *et al*, 2017), we listed 42 hot spots of TP53, including R270H mutation associated with the MVA pathway, and then verified their presence in tested cell lines.” is included at “Sanger sequencing” subsection of page 14 in Materials and Methods of the revised manuscript.

References

- Rivlin N, Brosh R, Oren M, Rotter V (2011) Mutations in the p53 Tumor Suppressor Gene: Important Milestones at the Various Steps of Tumorigenesis. *Genes Cancer* 2: 466–474
- Freed-Pastor WA, Mizuno H, Zhao X, Langerød A, Moon SH, Rodriguez-Barrueco R, Barsotti A, Chicas A, Li W, Polotskaia A *et al* (2012) Mutant p53 disrupts mammary tissue architecture via the mevalonate pathway. *Cell* 148: 244–258
- Sorrentino G, Ruggeri N, Specchia V, Cordenonsi M, Mano M, Dupont S, Manfrin A, Ingallina E, Sommaggio R, Piazza S *et al* (2014) Metabolic control of YAP and TAZ by the mevalonate pathway. *Nat Cell Biol* 16: 357–366
- Parrales A, Ranjan A, Iyer SV, Padhye S, Weir SJ, Roy A, Iwakuma T (2016) DNAJA1 controls the fate of misfolded mutant p53 through the mevalonate pathway. *Nat Cell Biol* 18(11): 1233–1243
- Turrell FK, Kerr EM, Gao M, Thorpe H, Doherty GJ, Cridge J, Shorthouse D, Speed A, Samarajiwa S, Hall BA *et al* (2017) Lung tumors with distinct p53 mutations respond similarly to p53 targeted therapy but exhibit genotype-specific statin sensitivity. *Genes Dev* 31: 1339–1353

4. There are multiple examples of where English grammar needs to be corrected.

Response: The manuscript has been edited by a professional English editing service.

Referee #2:

In the manuscript "Targeting YAP to overcome acquired resistance to ALK inhibitors in ALK-rearranged lung cancer" Yun et al. investigate mechanisms leading to resistance to ALK inhibitors in NSCLC lines and PDXs. The Authors discover that inhibition of the Mevalonate pathway sensitizes resistant cells to ALK inhibitors and link this to the activation of YAP. Loss of function and rescue experiments support a role of YAP in mediating resistance to ALK-inhibitors. Although YAP has already been linked to therapy resistance, its role in ALK-targeted therapy has not been addressed, thus the present work is novel and relevant. Overall the study is well designed and executed, there are only few points that need to be clarified.

Response: We thank the Reviewers for their insightful and positive comments and suggestions.

Our specific responses to the reviewers' comments are individually detailed below.

We hope that this revised text makes our manuscript more suitable for publication in *EMBO Molecular Medicine*.

Once again, we really appreciate that you have kindly edited and improved our manuscript.

Thank you for your comments and courtesy.

Major points

1. Figure 1E: is there a reason why FPP fails to rescue viability when used in combination with Cerivastatin? This is rather unexpected since GGPP (which is downstream FPP) is instead rescuing.

Response: Thank you for your insightful comments. The schematic of the mevalonate pathway (MVP) shown in the Appendix Fig S3 of our manuscript has a few steps omitted to simplify the pathway (Mullen *et al*, 2016; Iannelli *et al*, 2018).

The steps are as follows:

1) Mevalonate is phosphorylated and then decarboxylated to yield isopentenyl pyrophosphate (IPP), which can reversibly isomerize to dimethylallyl pyrophosphate (DMAPP).

2) Both DMAPP and IPP serve as substrates for FPP synthase which generates first geranyl pyrophosphate (GPP) and then farnesyl pyrophosphate (FPP).

3) Synthesis of geranylgeranyl pyrophosphate (GGPP) is catalyzed by GGPP synthase from conjugation of FPP with IPP.

Thus, supplemented FPP could not be sufficiently converted into GGPP because the generation of IPP is blocked upon cerivastatin treatment.

Moreover, since FPP also yields squalene by the action of squalene synthase (SQS), a portion of the supplemented FPP could be shunted towards cholesterol synthesis. For this reason, FPP may not have been able to rescue anti-proliferative effect of cerivastatin. This is consistent with our results showing that adding squalene had very little or no effect on cerivastatin-induced growth inhibition (Fig 1E).

Additionally, we have considered the following possible explanation as the major mechanism underlying YAP activation because cerivastatin exhibited anti-cancer effects by inhibiting YAP activation in our study. FPP and GGPP are essential substrates for farnesylation (via farnesyl transferase) and geranylgeranylation (via geranylgeranyl transferase), respectively, which is collectively referred to as protein prenylation (Wang & Casey, 2016). This process is required for membrane localization and activity of the RAS small guanosine triphosphatase (GTPase) superfamily, such as Ras, Rho, and Rab, which are closely involved in tumor progression. It is widely recognized that Ras proteins are typically farnesylated, whereas Rho and Rab proteins are geranylgeranylated. Moreover, the Rho GTPases have been recently identified as one of the key upstream inputs for YAP activity by modulating the actin cytoskeleton (Sorrentino *et al*, 2014; Wang *et al*, 2014; Mi *et al*, 2015).

Given the important role of Rho in promoting YAP activity, GGPP as a Rho prenylation substrate may be the key metabolite of the mevalonate pathway that enables YAP activation in our study.

Indeed, we found that re-exposure to GGPP prevented the inhibitory effects of cerivastatin and atorvastatin on YAP activation (Fig 2A and Fig EV1E). Consistent with the rescue experiment results, geranylgeranyl transferase inhibitor GGTI-298 was able to mimic the effect of cerivastatin on transcriptional activity of YAP (Fig 2F).

We additionally tested the farnesyl transferase inhibitor FTI-277 to analyze whether farnesylation was involved in cerivastatin action. As shown in Appendix Fig S4 below, FTI-277 only weakly

increased p21 expression and we failed to detect c-PARP and c-Cas3 levels. Moreover, FTI-277 had no effect on YAP phosphorylation and its target gene CYR61 expression.

Appendix Figure S4. Effect of FTI-277 in CR cells.

Representative immunoblots of the indicated proteins in lysates of cells treated with FTI-277 for 24 h. Blots are representative of three independent experiments.

Taking all this into consideration, protein geranylgeranylation is responsible for the positive effect of the mevalonate pathway in the context of YAP regulation. For this reason, we believe that GGPP but not FPP is able to rescue the anti-cancer effect of statin in CR cells. In line with our findings, a number of studies have shown that rescue by FPP supplementation on statin effects was not as robust as GGPP (Sorrentino *et al*, 2014; 9. Alarcon *et al*, 2016; Yu *et al*, 2018; Liu *et al*, 2018). However, we did not include the description mentioned above in the revised manuscript due to space limitations. We hope you would understand our circumstances.

Instead, we added the following sentence to page 7 in the Results of the revised manuscript.

“However, farnesyl transferase inhibitor FTI-277 only weakly increased p21 expression and we failed to detect c-PARP and c-Cas3 levels (Appendix Fig S4).”

References

- Mullen PJ, Yu R, Longo J, Archer MC, Penn LZ (2016) The interplay between cell signalling and the mevalonate pathway in cancer. *Nat Rev Cancer* 16: 718–731
- Iannelli F, Lombardi R, Milone MR, Pucci B, De Rienzo S, Budillon A, Bruzzese F (2018) Targeting Mevalonate Pathway in Cancer Treatment: Repurposing of Statins. *Recent Pat Anticancer Drug Discov* 3: 184–200
- Wang M, Casey PJ (2016) Protein prenylation: unique fats make their mark on biology. *Nat Rev Mol Cell Biol* 17: 110–122
- Sorrentino G, Ruggeri N, Specchia V, Cordenonsi M, Mano M, Dupont S, Manfrin A, Ingallina E, Sommaggio R, Piazza S *et al* (2014) Metabolic control of YAP and TAZ by the mevalonate pathway. *Nat Cell Biol* 16: 357–366
- Wang Z, Wu Y, Wang H, Zhang Y, Mei L, Fang X, Zhang X, Zhang F, Chen H, Liu Y *et al* (2014) Interplay of mevalonate and Hippo pathways regulates RHAMM transcription via YAP to modulate breast cancer cell motility. *Proc Natl Acad Sci U S A* 111: E89–E98
- Mi W, Lin Q, Childress C, Sudol M, Robishaw J, Berlot CH, Shabahang M, Yang W (2015) Geranylgeranylation signals to the Hippo pathway for breast cancer cell proliferation and migration. *Oncogene* 34: 3095–3106
- Alarcon VB, Marikawa Y (2016) Statins inhibit blastocyst formation by preventing geranylgeranylation. *Mol Hum Reprod* 22: 350–363
- Yu R, Longo J, van Leeuwen JE, Mullen PJ, Ba-Alawi W, Haibe-Kains B, Penn LZ (2018) Statin-Induced Cancer Cell Death Can Be Mechanistically Uncoupled from Prenylation of RAS Family Proteins. *Cancer Res* 78: 1347–1357

8. Liu BS, Xia HW, Zhou S, Liu Q, Tang QL, Bi NX, Zhou JT, Gong QY, Nie YZ, Bi F (2018) Inhibition of YAP reverses primary resistance to EGFR inhibitors in colorectal cancer cells. *Oncol Rep* 40: 2171-2182

2. Regarding the potential role of TAZ in mediating chemosensitivity/resistance, based on the transcriptional effect of TAZ knockdown, the Authors conclude that YAP (but not TAZ) is the target of Cerivastatin. The fact that loss of TAZ does not recapitulate all the transcriptional changes observed when treating with Cerivastatin does not imply that TAZ is not relevant. Authors need to silence TAZ and assess whether this will sensitize resistant cells (CR and LR cells) to ALK inhibitors.

Response: We absolutely agree with your comment. We have already shown that TAZ knockdown had no inhibitory effect on the clonogenicity of resistant cells in the original figure 4A, but the result description was missing from the original manuscript. It was clearly our mistake. So, we added the description at 2nd paragraph on page 9 in Results of the revised manuscript as follows:

“Although silencing of YAP alone markedly reduced the proliferation and clonogenicity of CR cells mainly due to cell cycle arrest at G0/G1 phase with induction of p21 expression, combined treatment of crizotinib and YAP knockdown resulted in a more prominent inhibitory effect (Fig 4A and Fig EV4). In contrast, TAZ silencing failed to attenuate the clonogenicity of CR cells (Fig 4A). Similar results were obtained with ceritinib-acquired resistant cells (LR pool and LR #6) displaying higher expression of YAP and YAP-target genes compared with that of parental cells (Fig EV5 and Appendix Fig S9).”

Figure 4. Effect of YAP knockdown on cell growth and tumor growth *in vitro* and *in vivo*.
A Colony formation in the indicated cells treated with either DMSO or crizotinib 24 h after siRNA transfection.

Figure EV5. Effect of YAP/TAZ knockdown on clonogenicity of ceritinib resistant cells.

A Representative immunoblots of the indicated proteins in lysates of LR cells transiently transfected with either negative control siRNA (Con si), YAP siRNAs (two sets of siRNAs against YAP; YAP si#1 and YAP si#2), TAZ siRNAs (two sets of siRNAs against TAZ; TAZ si#1 and TAZ si#2), or combination of TAZ siRNAs with YAP si #2 (Y+T si #1 and Y+T si #2).

B Colony formation in the indicated cells treated with either DMSO or ceritinib 24 h after siRNA transfection.

In addition, this sentence “These results suggest that YAP may be an integral part of a signaling network related to anticancer activity of cerivastatin in CR cells.” was replaced by “Given that YAP and its paralogue transcriptional co-activator with PDZ-binding motif (TAZ) are known to be functionally redundant and similarly regulated by Hippo signaling (Zhang *et al*, 2009; Moroishi *et al*, 2015; Yu *et al*, 2015), we further determined the role of TAZ in our system. Unlike YAP, TAZ knockdown did not affect the expression of EGFR, AXL and TGFβR2, but robustly inhibited CYR61 and VIM expression and increased IGFBP3 expression (Fig EV2 and Appendix Fig S7). These results suggest that there are potential differences between YAP and TAZ in the context of ALK-TKI resistance.”

Finally, we included the following paragraph on page 13 of the Discussion in the revised manuscript.

“Growing evidence suggests that TAZ promotes resistance to various anti-cancer therapies including cytotoxic chemotherapy (Moroishi *et al*, 2015; Zhan *et al*, 2018; Liu *et al*, 2019). However, our results showed that TAZ knockdown had no inhibitory effect on the clonogenicity of CR and LR cells (Fig 4A and Fig EV5). Interestingly, TAZ knockdown resulted in a marked increase in tumor suppressor IGFBP3 expression and complete inhibition of EMT associated VIM expression, but YAP did not influence the expression of these genes (Appendix Fig S7). These results raise the possibility that TAZ may, at least in part, be associated with ALK-TKI resistance through distinct transcriptional events. Indeed, several studies have shown that YAP and TAZ regulate different downstream target genes by tissue-specific or cell type-specific transcription factor binding partners (Varelas *et al*, 2008; Zhang *et al*, 2009). Moreover, Mi *et al* (2015) have reported that YAP mainly

contributes to cell proliferation while TAZ appears to regulate migration in breast cancer. Therefore, it is necessary to further investigate the functional role of TAZ in ALK-TKI resistance.”

3. Page 10: "However, YAP suppression could not restore the sensitivity to crizotinib due to ALK-independent resistance mechanism in resistant cells (Fig. 4D)". This observation is unexpected and at odds with the Authors main claim. Did the Authors check the level of YAP knockdown at the end of the experiment? This unexpected result could be due to clonal selection of cells with high YAP levels.

Response: Thank you for your insightful and helpful comments. Accordingly, we additionally investigated YAP knockdown levels in tumor samples at the end of the experiment. YAP expression levels in YAP shRNA- cells was completely inhibited compared with control shRNA- cells (Fig 4B). On the other hand, in tumor samples, YAP expression in YAP shRNA tumors was lower than that of control shRNA tumors, but the YAP shRNA tumor itself exhibited a high basal expression of YAP.

Additional Figure. Effect of YAP knockdown on tumor growth in Con shRNA- or YAP shRNA- xenograft

A Tumor growth curve in Con shRNA- or YAP shRNA- stable CR pool cells-derived xenografts during treatment with crizotinib (50 mg/kg). (Kruskal-Wallis followed by Dunn's post hoc test: $*P < 0.05$ vs. vehicle. ns, not significant. $n = 6$.)

B Representative immunoblot of YAP protein and representative images of YAP IHC staining in Con shRNA- or YAP shRNA- vehicle tumors at the end of experiment of drug treatment. Scale bar, 20 μm

As you commented, regarding our data that tumor generation in the YAP shRNA group was much slower than that of the control shRNA group after cell inoculation (Fig 4C), tumors of the YAP shRNA group may eventually be caused by clones expressing YAP.

Therefore, these findings are not included in the revised manuscript due to a false experimental design.

To obtain results that could support our claim, we had to start drug treatment as soon as the cells were inoculated. It would be consistent with our claim if we obtain results showing that tumor generation in the crizotinib-treated YAP shRNA group is more delayed than that in the YAP shRNA group.

However, the above experiment could not be performed within the limited revision period. Instead, we performed additional experiments to test the antitumor efficacy of verteporfin, as a pharmacological inhibitor of YAP, on the CR pool xenograft. Figure 4D is replaced by this new data.

Figure 4. Effect of YAP knockdown on cell growth and tumor growth *in vitro* and *in vivo*.
 D Tumor growth curves of CR pool cells-derived xenografts during treatment with the indicated drugs. * $P < 0.05$, ** $P < 0.01$ vs. vehicle. ## $P < 0.01$ vs. crizotinib treatment. ns, not significant. $n = 6$

This sentence "However, YAP suppression could not restore the sensitivity to crizotinib due to ALK-independent resistance mechanism in resistant cells (Fig. 4D)." is replaced by "In line with results, a YAP pharmacological inhibitor VP treatment yielded superior tumor growth inhibition compared with vehicle in CR pool xenograft (Fig. 4D). VP in combination with crizotinib produced antitumor synergy compared with single agent treatment."

Minor points

1. Why is Cerivastatin treatment leading to p21 increase, is p21 repressed by YAP? Is p53 activated?

Response: Thank you for your insightful comments. Considering our observation that both cerivastatin treatment and YAP inhibition led to upregulation of p21 in resistant cells rather than parental cells (Fig 1F and Fig EV1B and EV4B and Appendix Fig S2B and Appendix Fig S9B), there is a possibility that GGPP-mediated YAP activity may affect p21 expression in resistant cells. However, Appendix Fig S7A showed that the basal expression levels of p21 are similarly low in both parental and CR cells. For this reason, 1) we could exclude that active YAP in resistant cells reduces basal expression of p21. (The protein signal for p21 was not included in the original Figure because of no observable difference in p21 expression levels between parental and CR cells. So, we have added it to Appendix Fig S7A of the revised Figure.)

Appendix Figure S7. Effect of YAP and TAZ inhibition on expression of YAP associated genes in CR pool cells.

A Representative immunoblots of the indicated proteins in basal lysates of CR cells compared with parental cells.

Moreover, p21 is generally a p53 target gene (Dergham *et al*, 1997) yet both parental and resistant cells employed in our study expressed mutant p53 (E285V) exhibiting oncogenic activity (Appendix Table S1) and a previous report showed that mutant forms of p53 cannot bind to the p21 promoter

(Zhu *et al*, 2015), indicating that 2) p21 upregulation by cerivastatin did not occur due to p53 activation.

Variant	Alteration	Parental	CR pool	CR #1	CR #3	Variant	Alteration	Parental	CR pool	CR #1	CR #3
A159D	c.476C>A					R175L	c.524G>T				
A159P	c.475G>C					R248G	c.742C>G				
A159V	c.476C>T					R248L	c.743G>T				
E285V	c.854A>T	detected	detected	detected	detected	R248P	c.743G>C				
G245A	c.734G>C					R248Q	c.743G>A				
G245C	733G>T					R248W	c.742C>T				
G245D	c.734G>A					R249M	c.746G>T				
G245F	c.733_734GG>TT					R249S	c.747G>T				
G245N	c.733_734GG>AA					R273C	c.817C>T				
G245R	c.733G>C					R273G	c.817C>G				
G245S	c.733G>A					R273H	c.818G>A				
G245V	c.734G>T					R273L	c.818G>T				
R158C	c.472C>T					R273P	c.818G>C				
R158G	c.472C>G					R273S	c.817C>A				
R158H	c.473G>A					R280K	c.839G>A				
R158L	c.473G>T					R282W	c.844C>T				
R158P	c.473G>C					V157D	c.470T>A				
R158S	c.472C>A					V157F	c.469G>T				
R175C	c.523C>T					V157G	c.470T>G				
R175G	c.523C>G					V157L	c.469G>C				
R175H	c.524G>A					Y220C	c.659A>G				

Appendix Table S1. TP53 mutational status in crizotinib-resistant cells compared with H3122 parental cells

Lo Sardo *et al* (2017) has reported that p21 is maintained at low levels in lung tumors compared with normal tissues, which is similar to our results. They also revealed that YAP/TAZ interference or cerivastatin treatment upregulates p21 expression through post-transcriptional regulation by inhibiting oncogenic microRNAs such as miR-25,93 and 106b. Another study has suggested that YAP inactivation induces cell cycle arrest at the G1 phase by down-regulating Skp2, causing p21 to accumulate (Jang *et al*, 2017).

Although we did not verify the above molecular mechanisms in our experimental system, we considered the possibility that these mechanisms may be associated with a relevant input inducing p21 upregulation upon YAP inhibition by cerivastatin.

We hope to examine the detailed mechanism for YAP-induced p21 upregulation in a future study. We hope you would understand our circumstances.

References

1. Dergham ST, Dugan MC, Joshi US, Chen YC, Du W, Smith DW, Arlauskas P, Crissman JD, Vaitkevicius VK, Sarkar FH (1997) The clinical significance of p21(WAF1/CIP-1) and p53 expression in pancreatic adenocarcinoma. *Cancer* 80: 372-381
2. Zhu J, Sammons MA, Donahue G, Dou Z, Vedadi M, Getlik M, Baryte-Lovejoy D, Al-awar R, Katona BW, Shilatifard A *et al* (2015) Gain-of-function p53 mutants co-opt chromatin pathways to drive cancer growth. *Nature* 525: 206-211
3. Lo Sardo F, Forcato M, Sacconi A, Capaci V, Zanconato F, Di Agostino S, Del Sal G, Pandolfi PP, Strano S *et al* (2017) MCM7 and its hosted miR-25, 93 and 106b cluster elicit YAP/TAZ oncogenic activity in lung cancer. *Carcinogenesis* 38(1):64-75
4. Jang W, Kim T, Koo JS, Kim SK, Lim DS (2017) Mechanical cue-induced YAP instructs Skp2-dependent cell cycle exit and oncogenic signaling. *EMBO J* 1: 2510-2528.

2. Figure 2B: Why is YAP upregulated when cells are treated with GGPP and cerivastatin?

Response: As shown in Figure 2A, we obviously observed that supplementation of GGPP led to a decrease in YAP phosphorylation without affecting total YAP expression upon cerivastatin treatment. However, as you pointed out, co-treatment with GGPP and cerivastatin reduced YAP phosphorylation accompanied by an increase in total YAP expression upon siRNA transfection (Fig 2B of our original Figure is now included in Appendix Fig S5). In several repeated experiments, we obtained similar results. Unfortunately, we do not know the mechanism behind this phenomenon, but we speculate that a certain mechanism may influence YAP expression upon transfection.

More importantly, the purpose of the experiment was to determine whether GGPP-mediated YAP activation is regulated through the Hippo pathway core component LATS. Appendix Fig S5 shows that GGPP addition was remarkably inhibited cerivastatin-induced elevation of YAP phosphorylation despite an increase in total YAP expression, indicating that it is largely independent of LATS kinase activity.

We sincerely apologize that we were unable to provide a satisfactory answer to your question.

3. Figure 2C,D: What are the values shown in the heatmaps?

Response: We apologize for missing the values and have accordingly clarified in Figure legends of revised manuscripts as follows. "Colors represent z-scores."

4. Figures 1, 3 and 4: please indicate the number of mice used in the experiments shown.

Response: As suggested, we have included the number of mice used in the Figure and Figure legends of the revised manuscript.

5. Page 11: "In xenograft models, following subcutaneous cell injection, tumor from control cell were mostly observed 2 weeks, but those from stable YAP- knockdown cells were started to appear about 1 month and consequently formed smaller tumors at the end of an experiment (Fig. 4B and 4C)". Authors need to rephrase this: "within 2 weeks", "appear in about"

Response: We appreciate your kind comments and have corrected these phrases accordingly.

6. Page 12: "Of note, 60% (3/5) of tumors showing stable disease (SD) or progressive disease (PD) as the best response to ALK-TKI had high staining intensity, whereas 90% (9/10) of partial response (PR) samples showed low to intermediate staining intensity in pre-treatment tumor biopsies (Fig. 6C)". Please specify if here Authors are referring to YAP staining.

Response: As mentioned in the text, we obtained tumor biopsies from 17 patients treated with an ALK inhibitor (crizotinib or ceritinib), and paired pre- and post-treatment samples were available for nine of these patients. (Table 1).

The quantitative evaluation of YAP staining has been described in the "Immunohistochemistry (IHC)" subsection in the Materials and Methods of the revised manuscript and the IHC score of YAP in each patient is presented in Table 1.

Figure 6B shows a comparison of the overall YAP staining score between pretreatment and progression groups, in which the intensity of overall YAP expression was not significantly different between groups.

Next, to determine whether the elevated expression of YAP is associated with poor prognosis, the origin patients of tumor biopsies were classified into PR and SD/PD groups according to the best response to ALK-TKI, and then YAP staining intensity was compared between the pretreatment tumor biopsies of the two groups (Fig 6C).

Of the 10 PR patients, 9 (90%) showed a YAP staining score of 1 (low) to 2 (intermediate) in pretreatment tumor biopsies. On the other hand, 6 of 17 patients showed stable disease (SD) or progressive disease (PD) and one of them had no pretreatment tumor biopsies. Of the 5 SD/PD patients, 3 (60%) showed a YAP staining score of 3 (high) in pretreatment tumor biopsies.

However, the original text appears to be insufficient regarding explaining the results. Therefore, we revised the results description in the "Clinical implications of nuclear YAP expression in tumor biopsies from patients with ALK rearrangement."

Referee #3:

The manuscript by Yun and coworkers investigates the role of YAP signaling to treat NSCLC that have become resistant to ALK inhibitor therapy. This group has observed that some NSCLC which are treated with ALK inhibitor therapy, become resistant to therapy. While some have ALK dependent resistance, this group try's to find the cause and treatment for the cancers that become resistant independent of ALK signaling. using a ALK resistant cell line generated by this group, the group performed a screen of a 640 FDA approved drug library. They found that cerivastatin which targets the mevalonate pathway was the most selective. They then demonstrate that cerivastatin targets the YAP pathway and go to evaluate the role of the YAP pathway in the regulation of lung cancer cell growth in vitro and in patient derived xenograph models. They also demonstrate that nuclear localization of YAP is a clinically relevant alteration in human lung cancer. All in all this is a thorough and well-designed manuscript. They conducted a screen for a new therapy to treat ALK

inhibitor resistant lung cancer and have identified a novel pathway to target customized therapy for the treatment of lung cancer. All in all this is a novel and important study.

Response: We really appreciate your positive and encouraging comments and hope that our revised manuscript is now suitable for publication in *EMBO Molecular Medicine*. Thank you again for your comments and courtesy.

2nd Editorial Decision

4 September 2019

Thank you for the submission of your revised manuscript to EMBO Molecular Medicine. We have now received the enclosed reports from the referees that were asked to re-assess it. As you will see the reviewers are now globally supportive and I am pleased to inform you that we will be able to accept your manuscript pending minor editorial amendments.

Please submit your revised manuscript within two weeks. I look forward to seeing a revised form of your manuscript as soon as possible.

***** Reviewer's comments *****

Referee #1 (Remarks for Author):

The authors have presented a novel approach with mechanistic underpinnings of how to improve currently available targeted therapy for EML4-ALK translocated lung cancers. The authors have responded to all of the reviewers' comments including providing considerable additional data.

Referee #2 (Remarks for Author):

The Authors addressed most of the issues raised in my previous review, except the ones that I am reporting below.

Concerning my major point # 2 relative to TAZ role in chemoresistance:

- (i) I do not see a more prominent effect of YAP silencing in CR cells when these cells are treated with cerivastatin (figure 4a). (See also comment on figure 4a, reported below)
- (ii) siControl in the TAZ silenced cells panel displays poor growth, thus it is difficult to make any conclusion regarding TAZ silencing and chemoresistance, since the control itself shows poor growth (see colony assay in figure 4a).
- (iii) in figure EV5, siTAZ #1 in LR-pool causes a dramatic decrease of YAP, yet there is no effect on cells growth judging from the colony assay shown in panel B. This is inconsistent with the major conclusion of the manuscript, since this decrease of YAP should affect LR cells growth both in untreated and in ceritinib treated cells (as shown in the same figure when siYAP is used). I do not see any good explanation for this inconsistency, and would like to have a feedback from the Authors on this issue.

This notwithstanding, the overall data concerning TAZ looks preliminary and contradictory; for the sake of clarity and to avoid unnecessary confusion this data could be omitted from the manuscript.

Concerning my major point # 3:

The evidence of counter selection of tumor cells with higher YAP levels in the Sh-YAP xenograft is in line with the Authors claims, and could also be included a supplemental data since will contribute to reinforce the Authors main claim. On the other hand, the verteporfin (VP) experiment now shown in figure 4D is not very informative since VP alone is extremely effective in blocking tumor growth, thus making impossible to evaluate any combinatorial effect with crizotinib (see below, "additional remarks (b)" for more on this issue).

I also have some additional remarks,

- (a) Figure 3b: it is not clear what is statistically significant and what are the comparisons that had been evaluated. Also, while the result of the xenograft of the YAPS127A xenograft are displayed in

the figure, they are not sufficiently commented in the text: it is unclear whether combined cerivas+Crizo has a statistically significant increase in efficacy compared to cerivas alone.

(b) Figure 4 and 5 and relative text. Authors claim that "Although silencing of YAP alone markedly reduced the proliferation and clonogenicity of CR cells mainly due to cell cycle arrest at G0/G1 phase with induction of p21 expression, combined treatment of crizotinib and YAP knockdown resulted in a more prominent inhibitory effect (Fig 4A and B and Fig EV4)." I do not see evidences of combinatorial efficacy of targeting ALK by crizotinib and deactivating YAP either by silencing (figure4a) or by Verteporfin treatment (figure4d). In both instances, decommissioning of YAP leads to a strong decrease in cell growth (figure4a) or tumor growth (figure4d) which does not seem to be further increased by crizotinib. Regarding figure4a, the Authors should clarify if they have a quantitative data with available statistical evaluation. Concerning figure 4d, Authors should indicate in a clear way if there is a statistically significant difference between VP and VP+crizo. In any case, the two curves are very close to each other thus suggesting that the benefit of the combinatorial treatment is minimal. By the same token, it is unclear if the data shown in figure 5a (histogram graph) has been statistically evaluated in order to support the claim that combination of crizo+VP or crizo+cerivas shows increased efficacy. All of the above is crucial since, unless clarified, the conclusions of this paragraph ("Inhibition of YAP enhances tumor sensitivity to ALK-TKIs in mouse xenografts, patient-derived xenograft models and EML4-ALK transgenic mice ") do not seem to be supported by the data shown. In the present form, the data seem to point to an epistatic effect of YAP inhibition, rather than a true combinatorial therapeutic effect.

Minor point:

figure 4a, I believe that "cerit" in the figure label should be replaced with "crizo".

Referee #3 (Remarks for Author):

The authors have addressed the issues of the initial review.

2nd Revision - authors' response

20 September 2019

Referee #2 (Remarks for Author):

The Authors addressed most of the issues raised in my previous review, except the ones that I am reporting below.

Concerning my major point # 2 relative to TAZ role in chemoresistance:

(i) I do not see a more prominent effect of YAP silencing in CR cells when these cells are treated with cerivastatin (figure 4a). (See also comment on figure 4a, reported below)

Response: We absolutely agree with your comment on the combinatorial efficacy of YAP silencing and crizotinib in CR cells. (We understand that "cerivastatin" you mentioned above is "crizotinib".) We believe that the combination therapy with crizotinib seems relatively insignificant due to excellent anti-proliferative effect of YAP depletion against resistant cells. Thus, we have added quantitative graphs for colony formation in the revised Figure 4A. Although statistical significance was not strong as expected, the significance of $p < 0.05$ was achieved.

If acquired resistance to targeted therapy is due to bypass signal activity under the normal activity of driver oncogene, combination therapy with existing targeted therapy and inhibitors targeting resistance-inducing factor could provide superior antitumor synergy. However, our resistant cells exhibited lower total expression and phosphorylation of ALK compared with parental cells.

Moreover, other ALK inhibitors and ALK depletion failed to influence the survival of resistant cells. Based on these results, we did not expect potent antitumor synergy with combined therapy of ALK inhibitors and YAP inhibition in resistant cells with low ALK activity, but thought that improved antitumor effect in combination treatment compared to each single treatment was meaningful.

However, as we fully accept the reviewer's opinion in this matter, we have replaced the following statement "YAP silencing markedly reduced the proliferation and clonogenicity of CR cells mainly due to cell cycle arrest at G0/G1 phase with induction of p21 expression, which was slightly enhanced in co-treatment with crizotinib (Fig 4A and B and Fig EV4)."

(ii) siControl in the TAZ silenced cells panel displays poor growth, thus it is difficult to make any conclusion regarding TAZ silencing and chemoresistance, since the control itself shows poor growth (see colony assay in figure 4a).

Response: First, because there is a control siRNA for target-specific siRNA recommended by the manufacturer, we used different control siRNAs (Silencer™ Select Negative Control in Thermo Fisher Scientific and Acell Non-targeting pool in GE Dharamacon) for YAP and TAZ siRNAs, respectively.

To diminish any toxic effects from the transfection itself, we performed colony formation assay under optimal conditions to repress expression of target genes without affecting the cell viability of resistant cells. Optimal conditions were determined by changing the cell density, concentrations of siRNAs for negative control and target genes, and the batch of transfection reagent.

In concentration conditions, control siRNAs must be treated at the same concentration as the target-specific siRNA, so there was a difference in the treatment concentration. (The treatment concentrations of YAP and TAZ siRNAs were 20 nM and 50 nM, respectively, all of which are lower than the recommended concentration).

To assess the validity of siRNA experiment on the cell viability, we tested three types of control including transfection controls, untreated control, and negative control in all siRNA experiments. A fluorescent oligonucleotide (BLOCK-iT Fluorescent Oligo, Invitrogen) was used as a control for transfection to monitor the efficiency of siRNA transfection into the cells. Untreated control cultured without any siRNA treatment was used to determine the baseline levels of cell viability and target gene expression. Negative controls (control siRNAs) were used to distinguish sequence-specific silencing from non-specific effects. When negative control induced over $\pm 20\%$ change in cell viability compared to untreated control, indicating off-target effects, we reassessed usability of the data generated.

In this regard, all control siRNAs used met the optimal condition within the 10% range.

We deeply apologize for no data because this technical process was done to determine experimental conditions. We sincerely hope you would understand our circumstances.

(iii) in figure EV5, siTAz #1 in LR-pool causes a dramatic decrease of YAP, yet there is no effect on cells growth judging from the colony assay shown in panel B. This is inconsistent with the major conclusion of the manuscript, since this decrease of YAP should affect LR cells growth both in untreated and in ceritinib treated cells (as shown in the same figure when siYAP is used). I do not see any good explanation for this inconsistency, and would like to have a feedback from the Authors on this issue.

Response: Since YAP and TAZ share highly similar protein sequence and structure (60%), siRNA # 1 sequence for TAZ may be the portion where overlaps with YAP sequence. Although TAZ siRNA # 1 influenced the expression of YAP in CR cells as well as in LR cells, it did not block it as completely as YAP siRNA. This is consistent with the results showing that TAZ siRNA # 1 did not inhibit cell survival as well as expression levels of EGFR, AXL and TGFBR2 as YAP targets.

The bands are a technical error of exposure conditions on the blot following as:

1. Because the expression levels of TAZ is much higher than that of YAP in our cells, the detection time on the blot was very short in order to obtain a clear band for TAZ with an antibody that simultaneously detects YAP/TAZ. Under these conditions, YAP expression that was partially suppressed seems to appear to be completely suppressed due to relatively short exposure.
2. On the other hand, YAP expression by YAP siRNA was not observed at all, no matter how long the detection time was.

This notwithstanding, the overall data concerning TAZ looks preliminary and contradictory; for the sake of clarity and to avoid unnecessary confusion this data could be omitted from the manuscript.

Response: First of all, we sincerely apologize that we were unable to provide a satisfactory answer to your initial comments.

The functional redundancy between YAP and TAZ is already well known and we also demonstrated that TAZ inhibition shows excellent inhibitory effect on EMT marker VIM and increased expression of the tumor suppressor gene IGFBP3.

Thus, even though YAP has a stronger effect on cell survival than TAZ, we thought that TAZ could also be an important factor in other contexts of resistance (such as metastasis, for example).

However, as we fully agree with your opinions that the data on TAZ cause unnecessary confusion in the conclusion of manuscript, we have removed the TAZ results from the Figures.

Instead, we have included the TAZ results in the revised Appendix figure 9 and 10 in accordance with the editor's recommendation to keep it as an Appendix figure.

Concerning my major point # 3: The evidence of counter selection of tumor cells with higher YAP levels in the Sh-YAP xenograft is in line with the Authors claims, and could also be included a supplemental data since will contribute to reinforce the Authors main claim.

Response: Thank you for your insightful comments. Accordingly, we have included the results for Sh-YAP xenograft in the revised Appendix figure 11.

On the other hand, the verteporfin (VP) experiment now shown in figure 4D is not very informative since VP alone is extremely effective in blocking tumor growth, thus making impossible to evaluate any combinatorial effect with crizotinib (see below, "additional remarks (b)" for more on this issue).

Response: We will be happy to answer further in the "additional remarks (b)" section below.

I also have some additional remarks,

(a) Figure 3b: it is not clear what is statistically significant and what are the comparisons that had been evaluated.

Response: We sincerely acknowledge your comments. Thus, we have clearly indicated the statistical comparison and exact P-values are included in Appendix Table S5.

Also, while the result of the xenograft of the YAPS127A xenograft are displayed in the figure, they are not sufficiently commented in the text: it is unclear whether combined cerivas+Crizo has a statistically significant increase in efficacy compared to cerivas alone.

Response: An additional description for YAP-S127A xenograft in Figure 3b is given in the following sentence:

“Interestingly, YAP-S127A xenograft was less responsive to cerivastatin compared with YAP-WT xenograft. Moreover, combining of cerivastatin with crizotinib was comparable the effect of single agent treatment. These results may imply a limited inhibitory effect of cerivastatin on artificial YAP activity because YAP-S127A is not due to GGPP-mediated YAP activity. In line with these results, upon combination of cerivastatin and crizotinib, overall expression and nuclear localization of YAP were completely suppressed in tumor sections of YAP-WT, but nuclear YAP subpopulation was still detected in that of YAP-S127A (Fig 3C). Taken together, although both YAP overexpression and activation are responsible for resistance to crizotinib, transcriptional activity of YAP may be more aggressive and play a critical role in tumor progression.” This is included in the revised manuscript. Detailed description of the comments is below.

To verify whether YAP could confer resistance to crizotinib, we evaluated the sensitivity of crizotinib in tumor xenografts from pLVX cells, YAP-WT cells overexpressing wild-type YAP, and YAP-S127A cells overexpressing constitutively active YAP.

Figure 3B showed that the activity of crizotinib is markedly reduced in YAP-WT and YAP-S127A xenografts compared with the pLVX xenografts. Notably, combination of cerivastatin and crizotinib resulted in a synergistic effect in YAP-WT xenografts which was directly constructed from parent cells with high ALK activity (ALK-dependent). We believe that this result may be due to the fact that ectopic overexpressed cells are still dependent on ALK activity. These results support our argument that the combination of YAP inhibition and crizotinib showed little difference from the efficacy of YAP inhibition alone due to low ALK activity in resistant cells (ALK-independent).

On the other hand, cerivastatin yielded significant tumor growth inhibition in YAP-S127A xenograft, but was less effective compared with that of YAP-WT xenograft (Figure 3B). Moreover, combining cerivastatin with crizotinib was comparable to the effect of single agent treatment.

These results may be due to the following reasons.

- 1) Given that Figure 3A and the results of tumor growth comparison of vehicle groups of YAP-WT and YAP-S127A xenografts in Figure 3B, YAP-S127A tumors may be a much more aggressive than YAP-WT tumors, indicating the critical role of YAP transcription activity in tumor progression.
- 2) Since YAP-S127A is not due to GGPP mediated YAP activity, it may imply a limited inhibitory effect of statin on artificial YAP activity. Consistent with these results, upon combination of cerivastatin and crizotinib, overall expression and nuclear localization of YAP were completely suppressed in tumor sections of YAP-WT, but nuclear YAP subpopulation was still detected in YAP-S127A.

Indeed, among several upstream signaling pathway of YAP activity, statins are well known to impair YAP activation by inhibiting the geranylgeranylation (Mullen et al, 2016; Wang et al, 2014; Sorrentino et al, 2014; Mi et al, 2015). VP also inhibits the oncogenic function of YAP by destroying the YAP-TEAD complex. As such, there are no small molecule or antibody drugs that

can directly target YAP alone to this date. Therefore, these results provide a rationale of drug development by directly targeting YAP.

(b) Figure 4 and 5 and relative text. Authors claim that "Although silencing of YAP alone markedly reduced the proliferation and clonogenicity of CR cells mainly due to cell cycle arrest at G0/G1 phase with induction of p21 expression, combined treatment of crizotinib and YAP knockdown resulted in a more prominent inhibitory effect (Fig 4A and B and Fig EV4)." I do not see evidences of combinatorial efficacy of targeting ALK by crizotinib and deactivating YAP either by silencing (figure4a) or by Verteporfin treatment (figure4d). In both instances, decommissioning of YAP leads to a strong decrease in cell growth (figure4a) or tumor growth (figure4d) which does not seem to be further increased by crizotinib. Regarding figure4a, the Authors should clarify if they have a quantitative data with available statistical evaluation. Concerning figure 4d, Authors should indicate in a clear way if there is a statistically significant difference between VP and VP+crizo. In any case, the two curves are very close to each other thus suggesting that the benefit of the combinatorial treatment is minimal.

Response: We sincerely hope that this issue is also addressed in the same context as the answer to (i) in major point # 2 above. As suggested, we have added quantitative graphs for colony formation in the revised Figure 4A. The significance of $p < 0.05$ was achieved. In addition, concerning sentence was replaced by the following statements "YAP silencing markedly reduced the proliferation and clonogenicity of CR cells mainly due to cell cycle arrest at G0/G1 phase with induction of p21 expression, which was slightly enhanced in co-treatment with crizotinib (Fig 4A and B and Fig EV4)."

This sentence "VP in combination with crizotinib produced antitumor synergy compared with single agent treatment." has been removed in the revised manuscript.

By the same token, it is unclear if the data shown in figure 5a (histogram graph) has been statistically evaluated in order to support the claim that combination of crizo+VP or crizo+cerivas shows increased efficacy.

Response: Apologies for the mistake. Exact P-values are included in Appendix Table S5.

All of the above is crucial since, unless clarified, the conclusions of this paragraph ("Inhibition of YAP enhances tumor sensitivity to ALK-TKIs in mouse xenografts, patient-derived xenograft models and EML4-ALK transgenic mice ") do not seem to be supported by the data shown. In the present form, the data seem to point to an epistatic effect of YAP inhibition, rather than a true combinatorial therapeutic effect.

Response: As we agree with your kind comments, we have toned down our statements throughout the manuscript and the concerning sentence was replaced by the following statements "Inhibition of YAP overcomes resistance to ALK-TKIs in mouse xenografts, patient-derived xenograft models and EML4-ALK transgenic mice"

Minor point:

figure 4a, I believe that "cerit" in the figure label should be replaced with "crizo".

Response: Thank you for pointing out this typo. It has been corrected.

Corresponding Author Name: Byoung Chul Cho

Manuscript Number: EMM-2019-10581-V3